# Endothelin B receptor inhibition rescues aging-dependent neuronal regenerative decline

Rui Feng[1†], Sarah F Rosen[1†], Irshad Ansari[1], Sebastian John[1], Michael B Thomsen[2], Oshri Avraham[1‡], Cedric G Geoffroy[3], Valeria Cavalli[1,4,5]*

[1]Department of Neuroscience, Washington University School of Medicine, St Louis, United States; [2]CS27 Bioinformatics, Springboro, United States; [3]Department of Neuroscience & Experimental Therapeutics, Texas A&M Health Science Center, College Station, United States; [4]Center of Regenerative Medicine, Washington University School of Medicine, St Louis, United States; [5]Hope Center for Neurological Disorders, Washington University School of Medicine, St Louis, United States

*For correspondence: cavalli@wustl.edu

[†]These authors contributed equally to this work

Present address: [‡]Department of Cellular Biology, University of Georgia, Athens, United States

## eLife Assessment

This **important** study examines the role of endothelin signaling in nerve regeneration, providing **convincing** evidence that it functions as a default brake on axon regrowth. Inhibiting endothelin signaling with Bosentan promotes regeneration and counteracts the decline in regenerative potential caused by aging. Since Bosentan is an FDA-approved drug, these findings could have therapeutic value in clinical settings where peripheral nerve regeneration is not adequate or seriously impaired, as is often the case in older individuals.

**Abstract** Peripheral sensory neurons regenerate their axons after injury to regain function, but this ability declines with age. The mechanisms behind this decline are not fully understood. While excessive production of endothelin 1 (ET-1), a potent vasoconstrictor, is linked to many diseases that increase with age, the role of ET-1 and its receptors in axon regeneration is unknown. Using single-cell RNA sequencing, we show that satellite glial cells (SGCs), which completely envelop the sensory neuron soma residing in the dorsal root ganglia (DRG), express the endothelin B receptor (ETBR), while ET-1 is expressed by endothelial cells. Inhibition of ETBR ex vivo in DRG explant cultures improves axon growth in both adult and aged conditions. In vivo, treatment with the FDA-approved compound, Bosentan, improves axon regeneration and reverses the age-dependent decrease in axonal regenerative capacity. Single-nuclei RNA sequencing and electron microscopy analyses reveal a decreased abundance of SGCs in aged mice compared to adult mice. Additionally, the decreased expression of connexin 43 (Cx43) in SGCs in aged mice after nerve injury is partially rescued by Bosentan treatment. These results reveal that inhibiting ETBR function enhances axon regeneration and rescues the age-dependent decrease in axonal regenerative capacity, providing a potential avenue for future therapies.

## Introduction

Peripheral nerve injuries, resulting from traumatic injuries, lesions, degenerative diseases, and neuropathies, have a major impact on patient functioning and quality of life (*Maita et al., 2023*). While neurons in the peripheral nervous system regenerate their axons after nerve injury, functional recovery remains very limited because of the slow growth rate of axons and the often long distances that

growing axons face to reconnect with their initial targets (*Höke, 2006*). Unfortunately, the ability of injured neurons to regenerate their axons declines with age (*Pestronk et al., 1980*; *Vaughan, 1992*; *Verdú et al., 2000*), contributing to increases in healthcare costs and the risk of long-term disability (*Pestronk et al., 1980*; *Vaughan, 1992*; *Verdú et al., 2000*; *Yun, 2015*). Thus, the discovery of therapeutics that promote axon regeneration and counter age-related decline in regenerative capacity is vital.

Primary sensory neurons, with their cell soma located in the dorsal root ganglia (DRG), convey sensory information from peripheral tissue to the brain via the spinal cord and represent a useful model to identify the molecular and cellular mechanisms that promote axon regeneration. After injury, successful axonal regeneration of sensory neurons requires activation of neuronal-intrinsic epigenetic, transcriptional, and translational programs (*He and Jin, 2016*; *Mahar and Cavalli, 2018*; *Rishal and Fainzilber, 2014*; *Tedeschi and Bradke, 2017*). Additionally, non-neuronal cells at the site of injury and in the DRG play an important role in axon regeneration after injury. Satellite glial cells (SGCs) surrounding the cell soma (*Avraham et al., 2020*; *Avraham et al., 2021*; *Jager et al., 2020*) and Schwann cells surrounding the axon (*Jessen and Mirsky, 2016*) undergo changes in transcriptional states that support axon regeneration. Macrophages recruited to the nerve (*Ydens et al., 2020*; *Zigmond and Echevarria, 2019*), and macrophages proliferating in the DRG (*Feng et al., 2023*) also contribute to axon regeneration after injury.

During aging, decreases in the efficiency of the axonal transport system (*Andrews et al., 2016*; *Black and Lasek, 1979*), mitochondrial function (*Sutherland et al., 2021*), and epigenetic and transcriptional mechanisms (*Li et al., 2010*) limit the regenerative capacity of neurons. Increases in pro-inflammatory cytokines and macrophage infiltration into the peripheral nerve (*Büttner et al., 2018*), as well as an increased number of T cells in the DRG, have been shown to contribute to the limited regenerative capacity of neurons in aged mice (*Zhou et al., 2022*). Additionally, age-related decline in the de-differentiation, activation, and senescence of Schwann cells hampers axonal re-growth in damaged peripheral nerves (*Fuentes-Flores et al., 2023*; *Kang and Lichtman, 2013*; *Painter et al., 2014*). While changes in SGC morphology (*Pannese et al., 1996*), as well as changes in SGC-neuronal coupling have been reported with age (*Huang et al., 2006*; *Procacci et al., 2008*), the contribution of SGCs to age-dependent decreases in axonal regenerative capacity has not been investigated.

Endothelin B receptor (ETBR), a G-protein-coupled receptor, is expressed in SGCs (*Mapps et al., 2022*; *Pomonis et al., 2001*). One of the ligands for ETBR, Endothelin- 1 (ET-1), is a potent vaso-constrictor secreted by endothelial cells and is the predominant isoform of endothelin in the human cardiovascular system (*Lüscher and Barton, 2000*; *Yanagisawa et al., 1988*). Previous studies in trigeminal and nodose ganglia have shown that ET-1 activates SGCs through ETBR and reduces gap junction coupling in vitro (*Feldman-Goriachnik and Hanani, 2017*). Studies in astrocytes, which share many functional and molecular features with SGCs (*Avraham et al., 2022*; *Avraham et al., 2020*; *Hanani and Verkhratsky, 2021*), have shown that ETBR inhibition reduces reactive astrocytes and may be a beneficial target to modulate brain diseases (*Koyama, 2021*). Additional studies in astrocytes have shown that ET-1 signaling via ETBR inhibits the expression of Connexin 43 (Cx43) and reduces gap-junction-mediated coupling (*Blomstrand et al., 2004*; *Rozyczka et al., 2005*). Cx43 is the most highly expressed member of the connexin family in SGCs (*Avraham et al., 2022*; *Huang et al., 2005*) and its expression decreases during aging (*Procacci et al., 2008*). Notably, ET-1 production increases with age and is linked to many age-associated diseases (*Barton, 2014*; *Jankowich and Choudhary, 2020*; *Stauffer et al., 2008*). However, the function of ETBR in SGCs is not known, and whether endothelin signaling contributes to axon regeneration and age-dependent axon regenerative decline has not been examined.

First, to determine the role of endothelin signaling in axon regeneration, we performed unbiased single-cell RNA sequencing of lumbar DRG from mice 3 months of age (adult). Examination of endothelins and their receptors showed enrichment of *Edn1,* the gene encoding ET-1 in endothelial cells and *Ednrb*, the gene encoding ETBR, in SGCs. Inhibition of ETBR using the selective antagonist BQ788 in DRG explant cultures enhanced axonal outgrowth in adults and restored axonal regenerative capacity in aged conditions. Treatment of adult and aged mice with Bosentan, an FDA-approved ETBR/ETAR antagonist (*Clozel et al., 1994*), improved axon regeneration after peripheral nerve injury. To further examine the changes in SGCs induced by aging, we performed single-nuclei RNA-sequencing (snRNA-seq) and transmission electron microscopy of DRG cells from adult and

21-month-old mice (aged). We observed decreased abundance of SGCs in aged conditions, together with extensive transcriptional reprogramming in SGCs, reflecting heightened demands for structural integrity, cell junction remodeling, and glia–neuron interactions within the aged DRG microenvironment. Mechanistically, we found that Bosentan treatment increased expression of Cx43 in SGCs after injury in adult and aged mice. These results suggest that ETBR signaling limits axon regeneration after nerve injury and plays a role in age-related decline in regenerative capacity. Thus, ETBR antagonism may be a potential therapeutic avenue to enhance axon growth after nerve injury and to restore axon regenerative capacity that declines with age.

## Results

### *Ednrb* is highly expressed in SGCs

The dense vascularization of the DRG cell body area coupled with the high permeability of these capillaries in rodents and human (*Godel et al., 2016*; *Jimenez-Andrade et al., 2008*) suggests that neurons and their surrounding SGCs may be influenced by vascular-derived signals. Whole mount preparation of DRG from Lycopersicon Esculentum Lectin (LEL) injected *Fabp7*<sup>CreER</sup>::Ai14 mouse, which labels SGCs with tdTomato (*Avraham et al., 2020*) revealed the dense vascularization in the neuronal soma rich area of the DRG, as previously reported (*Jimenez-Andrade et al., 2008*; *Figure 1A*, *Figure 1—video 1*). Additionally, immunofluorescence staining of neurons and SGCs in the DRG from LEL-injected adult mouse demonstrated that blood vessels are closely juxtaposed to SGCs surrounding sensory neuron somas (*Figure 1B*).

To understand the role of endothelin intercellular signaling on axon regeneration, we first performed single cell RNA sequencing (scRNA-seq) on lumbar DRGs from adult female mice (3 months of age) using the Chromium Single Cell Gene Expression Solution (10 X Genomics), as described previously (*Avraham et al., 2022*; *Avraham et al., 2020*; *Avraham et al., 2021*). We sequenced a total of 16,764 cells from two female biological replicates with an average of 3579 genes and 12,090 transcripts detected per cell (*Figure 1C*, *Figure 1—figure supplement 1A and B*, see methods for filtering criteria). Following unsupervised clustering, we identified eight clusters corresponding to the principal cell types of the mouse DRG. Examination of the cluster-specific marker genes verified the presence of major cellular subtypes including SGCs (*Fabp7/Kcnj10*), myelinating Schwann cells (*Ncmap/Mag*), non-myelinating Schwann cells (*Scn7a/L1cam*), neurons (*Isl1/Prph*), endothelial cells (*Pecam1/Flt1*), fibroblasts (*Col1a1/Pdgfra*), mural cells (*Rgs5/Notch3*), and macrophages (*C1qa/Aif1*) (*Figure 1D*, *Figure 1—figure supplement 1C*, *Figure 1—source data 1*). We then examined the expression of endothelin and their receptors. Endothelin (*Edn1*) and endothelin-3 (*Edn3*) were expressed in endothelial cells and fibroblasts, respectively (*Figure 1D, G and H*). *Ednrb* was enriched in SGCs (*Figure 1D and F*), whereas *Ednra* was expressed mainly in mural cells (*Figure 1D and E*). *Ednrb* enrichment in SGCs is consistent with prior studies (*Mapps et al., 2022*; *Pomonis et al., 2001*; *Tasdemir-Yilmaz et al., 2021*). Using a published scRNAseq data set across species (*Bhuiyan et al., 2024*), we observed similar enrichment of *Ednrb* expression in SGCs in humans (*Figure 1—figure supplement 1D*). *Ednrb* mRNA in mouse SGCs was validated by RNA in situ hybridization, which showed *Ednrb* co-localized with the SGC marker *Fabp7* (*Avraham et al., 2020*; *Figure 1I*).

### Endothelin B receptor inhibition increases axonal growth in vitro and ex vivo

To investigate the role of ETBR signaling in axonal growth, we used different compounds to inhibit or activate endothelin receptors in dissociated L4-L5 DRG mixed cultures. In these DRG cultures, SGCs lose their adhesive contacts with neuronal soma and adhere to the coverslip as time progresses (*Belzer et al., 2010*; *Valtcheva et al., 2016*). Within 24 hr in culture, most SGCs are found close to neurons and display flat morphology (*Figure 2B*; *Belzer et al., 2010*). In these in vitro conditions, ET-1 or ET-3 may be released by endothelial cells or fibroblasts. DRG cells were plated and treated with an ETBR antagonist (BQ788), ETBR agonist (IRL1620), endothelin A receptor (ETAR) antagonist (BQ123), or vehicle for 24 hr (*Figure 2A*). Neurons were then labeled with antibodies against beta III tubulin (TUJ1), and axonal radial length was quantified. Antagonism of ETBR with BQ788 significantly increased axonal radial length compared to vehicle (*Figure 2C and D*). However, agonism of ETBR with IRL1620 or antagonism of ETAR with BQ123 had no significant effects on axonal radial

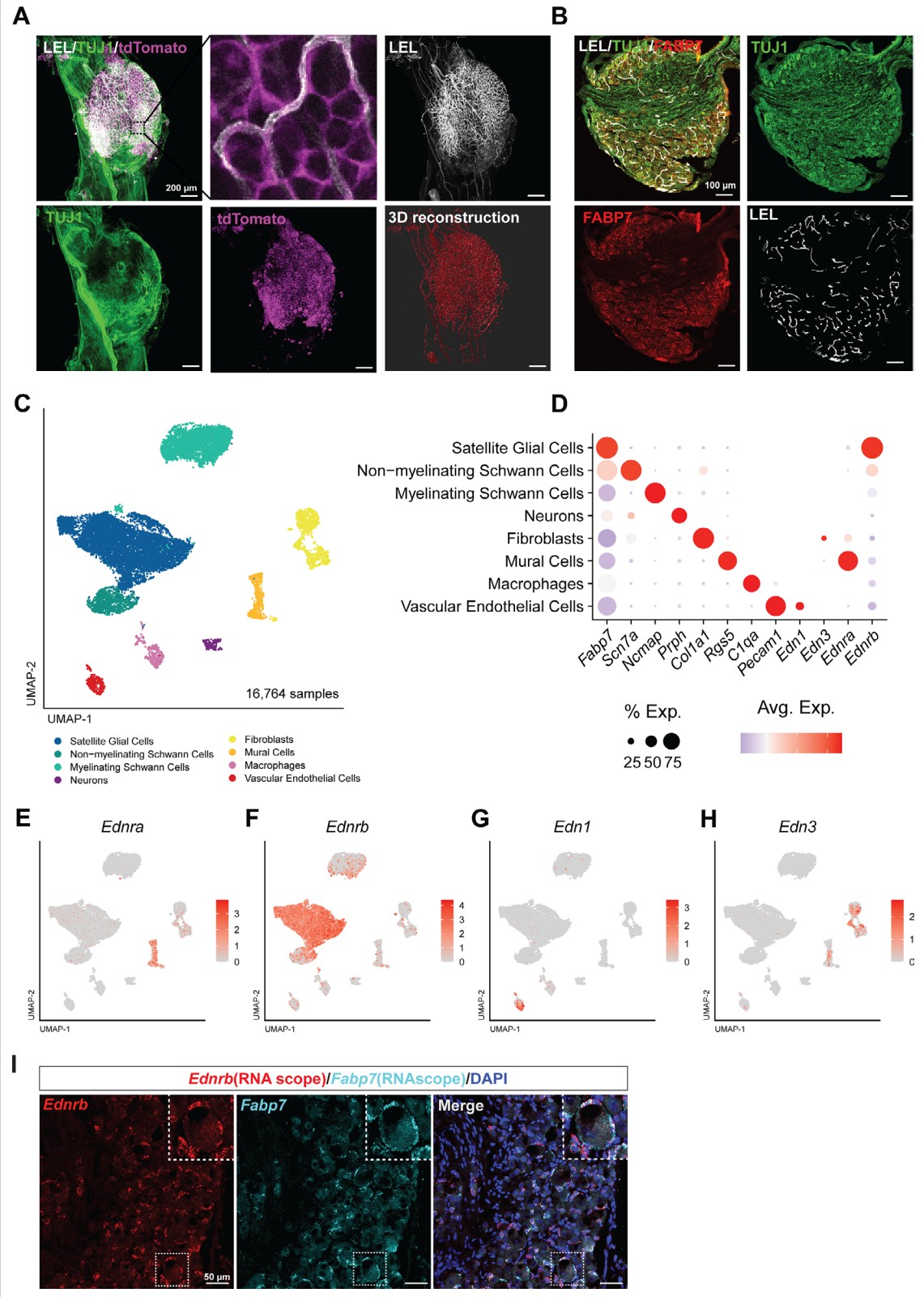

**Figure 1.** *Ednrb* is highly expressed in satellite glial cells. (**A**) Representative whole-mount-stained images of DRG from Lycopersicon Esculentum Lectin (LEL) injected Fabp7CreER::Ai14 mice, labeled for TUJ1 (green), tdTomato (magenta), and LEL (gray). 3D reconstruction of blood vessels via LEL labeling (scale bars, 200 μm). Note that TUJ1 antibody staining is limited by penetration of the antibody in the whole mount. (**B**) Representative images of sectioned DRG from Lycopersicon Esculentum Lectin (LEL) injected C57BL/6 mice, labeled for TUJ1 (green), Fabp7 (red), and LEL (cyan) (Scale bars,

*Figure 1 continued on next page*

*Figure 1 continued*

100 µm). (**C**) UMAP analysis of adult DRG 10 X sequencing data identified 8 cell clusters based on known marker genes. (**D**) Dot plot analysis showing the average gene expression (color coded) and number of expressing cells (dot size) for the marker genes. (**E–H**) UMAP overlay for expression of *Ednra* (**E**), *Ednrb* (**F**), *Edn1* (**G**), and *Edn3* (**H**). (**I**) Representative RNAScope in situ hybridization images showing *Ednrb* (red), *Fabp7* (cyan), and DAPI (blue) of L4 DRGs from 3-month-old mice (scale bars, 50 µm).

The online version of this article includes the following video, source data, and figure supplement(s) for figure 1:

**Source data 1.** Marker genes for 10x scRNA-seq analysis.

**Figure supplement 1.** Sample QC and DEG analysis for 10x scRNA-Seq data.

**Figure 1—video 1.** Z-stack video of whole mount preparation of DRG from Lycopersicon Esculentum Lectin (LEL) injected Fabp7CreER::Ai14 mouse, which labels SGCs with tdTomato, immunostained for TUJ1 (neurons).

https://elifesciences.org/articles/100217/figures#fig1video1

length compared to vehicle treatment (*Figure 2C and D*). These results indicate that ETBR signaling limits axonal growth in in vitro mixed culture conditions. Inhibition of ETBR may alter trophic factors produced by SGCs into the culture media, leading to axon growth-promoting conditions.

To further examine the role of ETBR antagonism on axonal growth, we used DRG explants in which L3-L5 DRGs from adult mice were seeded in matrigel and cultured for 7 days (*Feng et al., 2023*). Explants were then treated with vehicle or BQ788 at days in vitro (DIV) 1, and radial axon growth was assessed at DIV7 by measuring the length of 50 or more axons per explant (*Figure 2E*). Radial axon growth was significantly increased in explants treated with BQ788 compared to vehicle treatment (*Figure 2F and G*). To confirm that the in vivo morphology of SGCs enveloping neuronal somas was preserved, DRG explants were immunostained with TUJ1 for neurons and Fabp7 for SGCs at DIV 7 (*Figure 2H*). These results further indicate that ETBR signaling limits axon growth capacity in an ex vivo model.

## Bosentan treatment improves axon regeneration after nerve injury

To determine the potential role of ETBR in regulating nerve regeneration in vivo, we used our established in vivo regeneration assays (*Avraham et al., 2020*; *Cho and Cavalli, 2012*; *Cho et al., 2015*; *Cho et al., 2013*) in combination with the FDA-approved compounds that antagonize endothelin receptors, Bosentan and Ambrisentan. These compounds have been utilized to treat pulmonary arterial hypertension, and their safety and efficacy have been demonstrated through various clinical trials (*Chen et al., 2018*; *Galie, 2008*; *Peacock et al., 2015*; *Rubin et al., 2002*). Bosentan is an antagonist of both ETAR and ETBR (*Clozel et al., 1994*), while Ambrisentan is an antagonist selective for ETAR (*Humbert et al., 2004*; *Newman et al., 2007*). Adult mice were treated orally with vehicle, Bosentan, or Ambrisentan 2 hr prior to receiving a sciatic nerve crush (SNC) injury. Axon regeneration was assessed 24 hr after injury by measuring the intensity of SCG10 labeled axons, a marker for regenerating axons (*Shin et al., 2014*; *Figure 3A–E*). Since axon growth is relatively slow 24 hrs post injury (*Shin et al., 2012*), we first determined whether ETBR inhibition could accelerate axon growth in the first 24 hr (*Shin et al., 2012*; *Smith and Skene, 1997*). SCG10 intensity was measured distal to the crush site, which was determined according to the highest SCG10 intensity along the nerve (*Avraham et al., 2020*; *Cho et al., 2013*; *Feng et al., 2023*; *Shin et al., 2014*) and axon elongation was quantified by measurement of the 10 longest axons, as previously described (*Carlin et al., 2019*; *Figure 3B*). A 50% regeneration index was quantified by normalizing the average SCG10 intensity at distances away from the crush site to the SCG10 intensity at the crush site, to account for both the length and number of regenerating axons past the crush site and plotting the distance at which SCG10 intensity is half of that at the crush site (*Cho et al., 2013*; *Feng et al., 2023*; *Figure 3C* had significantly less axon regeneration compared). Mice that received Bosentan had significantly longer axons and a higher 50% regeneration index 1 day after SNC compared to mice that received vehicle, whereas mice that received Ambrisentan had no significant differences in axon regeneration compared to vehicle (*Figure 3B–D*). Quantification of the percentage of SCG10 intensity normalized to the crush site at several distances from the injury was also greater in Bosentan-treated mice compared to vehicle (*Figure 3E*). Similarly, at 3 days post-SNC, mice that received Bosentan 2 hr before SNC, and then every 24 hr, had improved axon regeneration compared to vehicle, whereas mice that received Ambrisentan had no significant differences in axon regeneration compared to

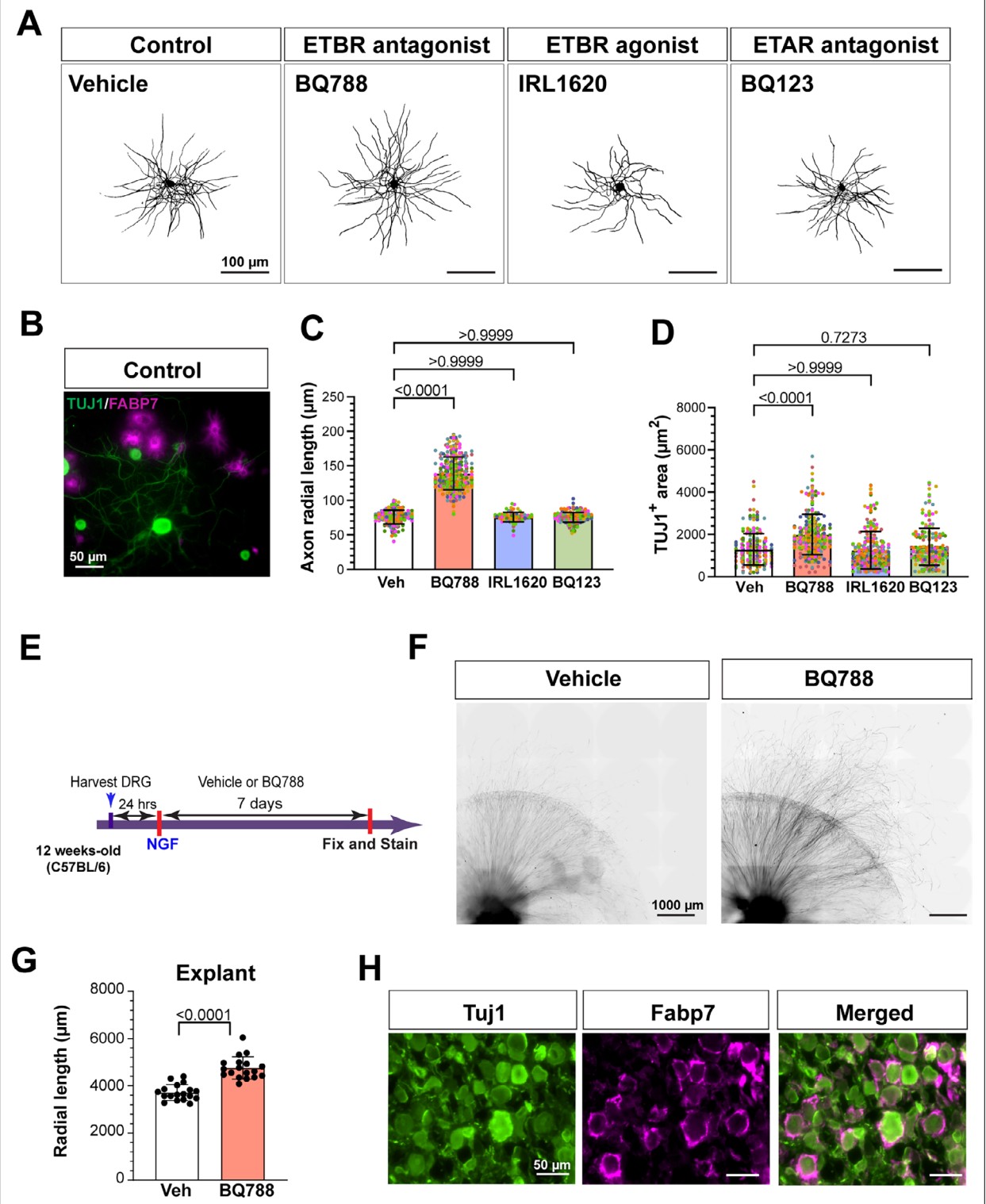

**Figure 2.** Endothelin B receptor inhibition increases axonal growth in vitro and ex vivo. (**A**) Representative images showing TUJ1 (black) immunostaining of neurons in DRG cultures (scale bars, 100 μm). (**B**) Representative image of TUJ1 (green) and Fabp7 (magenta) immunostaining of neurons and SGCs in control DRG cultures (scale bars, 50 μm). (**C, D**) Quantification of axonal radial length (**B**) and TUJ1+ area (**C**) per neuron. Different colors indicate biological replicates. N=246 (Veh; 8 replicates), 318 (BQ788; 8 replicates), 320 (IRL1620; 8 replicates), and 244 (BQ123; 8 replicates). Data presented as mean ± SD. (**E**) Scheme of drug treatment and DRG explant model. (**F**) Representative images of DRG explants 7 days after drug treatment, immunostained for TUJ1 (black) (scale bars, 1000 μm). (**G**) Quantification of radial length of the 35 longest axons from DRG explants from indicated groups N=36 explants from 6 individual mice (BQ788; 18 replicates, Veh; 18 replicates). The data are presented as mean ± SD. (**H**) Representative images of DRG explants immunostained for TUJ1 (green), FABP7 (magenta), and merged (scale bars, 50 μm).

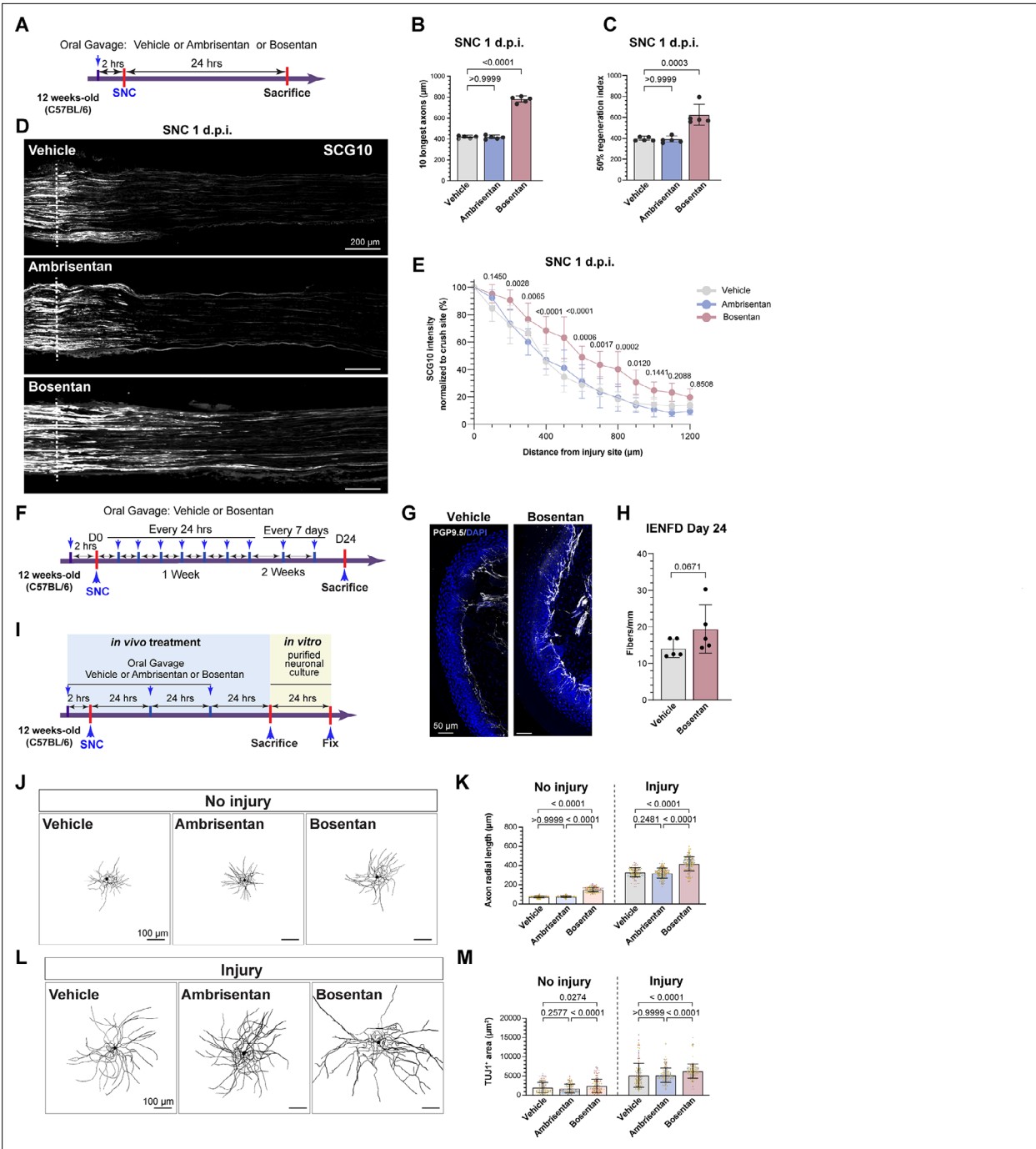

**Figure 3.** Bosentan treatment improves axon regeneration after peripheral nerve injury in adult mice. (**A**) Scheme of drug treatment and peripheral nerve injury model. (**B**) Quantification of the length of the 10 longest axons in indicated conditions. (**C**) Quantification of the regeneration index, calculated as the distance along the nerve where the SGC10 intensity is 50% of the SCG10 intensity at crush site. (**D**) Representative longitudinal sections of sciatic nerves 24 h after SNC, immunostained for SCG10, from mice with the indicated treatment. Dotted line indicates the crush site, determined as the maximal SGC10 intensity (scale bars, 200 μm). (**E**) Quantification of SCG10 intensity at the indicated distance normalized to the intensity at the crush site for each condition. N=5 mice/condition. (**F**) Scheme of long-term Bosentan treatment. (**G**) Representative images of hindpaw skin after long-term Bosentan treatment immunostained for PGP9.5 (white) and DAPI (blue) (scale bars, 50 μm). (**H**) Quantification of intraepidermal nerve fiber density (IENFD) 24 days after sciatic nerve crush from the indicated groups. N=5 mice/condition. (**I**) Scheme of adult DRG neuronal culture and treatments. (**J, L**) Representative images showing TUJ1 (black) immunostaining of neurons in DRG cultures (scale bars, 100 μm). (**K, M**) Quantification of axonal radial length (**K**) and total TUJ1⁺ area (**L**). Different colors represent different biological replicates. N (neuron number)=177 (vehicle; three biological replicates, naïve mice), 168 (Ambrisentan, three biological replicates, naïve mice), 183 (Bosentan; three biological replicates,

*Figure 3 continued on next page*

*Figure 3 continued*

naïve mice), 186 (vehicle; three biological replicates, injured mice), 204 (Ambrisentan, three biological replicates, injured mice), and 210 (Bosentan; three biological replicates, injured mice), respectively. The data are presented as mean ± SD.

The online version of this article includes the following source data and figure supplement(s) for figure 3:

**Figure supplement 1.** Bosentan treatment improves axon regeneration 3 days after peripheral nerve injury.

**Figure supplement 1—source data 1.** Original files for western blot analysis.

**Figure supplement 1—source data 2.** PDF file containing original western blots, indicating the relevant bands and treatments.

**Figure supplement 2.** Bosentan treatment improves axon regeneration after dorsal root crush injury.

vehicle (*Figure 3—figure supplement 1A–E*). Examination of *Edn1, Ednra, and Ednrb* mRNA by RT-qPCR and ETBR by western blot in non-treated mice showed no significant differences in mRNA or protein levels at 3 days post SNC compared to control (*Figure 3—figure supplement 1F–H*). These results suggest that, although ETBR levels do not change after injury, ETBR antagonism increases axon regeneration after peripheral nerve injury at both 1 and 3 days. We next sought to determine whether Bosentan treatment over the course of nerve regeneration improves target re-innervation. Regenerating axons of sciatic nerves extend to the epidermis and start to re-innervate the skin of the hind paw at ~2–3 weeks after injury. We administered Bosentan or vehicle 2 hours prior to SNC, then daily for 7 days, followed by once a week for 2 weeks (*Figure 3F*). Intraepidermal nerve fiber density (IENFD) in glabrous skin of the hind paw was assessed by quantification of PGP9.5 positive nerve fibers per millimeter of skin 24 days after SNC. Mice that received Bosentan had a trend towards increased IENFD compared to vehicle-treated mice (*Figure 3G and H*). These results suggest that Bosentan treatment may accelerate target re-innervation after injury.

We next tested if inhibition of ETBR influences the conditioning injury paradigm, in which a prior nerve injury increases the growth capacity of neurons (*Smith and Skene, 1997*). DRG neurons from mice that received Bosentan or Ambrisentan 2 hr before SNC, and then every 24 hr for 3 days, were cultured and axon growth capacity was quantified by measuring the axon radial length and TUJ1 positive area (*Figure 3J–M*). In uninjured condition, neurons extend short neurites, whereas a prior nerve injury leads to neurons growing longer neurites (*Figure 3J–M*), as expected (*Smith and Skene, 1997*). Bosentan, but not Ambrisentan, enhanced the axon radial length of DRG neurons cultured from uninjured adult mice, partially mimicking the conditioning injury, and also increased the growth capacity of injured neurons (*Figure 3J–M*). Since the selective ETAR antagonists, BQ123 and Ambrisentan, had no significant effects on axon regeneration after nerve injury (*Figures 2A–C and 3A–E*), our results indicate that ETBR signaling limits axon regenerative capacity.

To determine the effects of Bosentan treatment on axon regeneration in a model with lower regenerative capacity, axon growth after dorsal root crush (DRC) injury was assessed (*Figure 3—figure supplement 2A*). Following DRC injury, axon growth occurs at about half the rate of peripheral axons (*Oblinger and Lasek, 1984*; *Wujek and Lasek, 1983*) and our previous work has shown that manipulation of SGCs after DRC improves axon regeneration (*Avraham et al., 2021*). Axon regeneration at 3 days post-DRC was assessed by measuring SCG10 intensity, as described above. Mice that received Bosentan had significantly more axon regeneration along the dorsal root compared to vehicle (*Figure 3—figure supplement 2B–E*). Together, these results indicate that ETBR inhibition improves axon regeneration after both peripheral and central axon injury in DRG neurons.

## Inhibiting ETBR rescues aging-dependent axonal regenerative decline

It is known that elevated tissue or plasma concentrations of ET-1 occur with age (*Jankowich and Choudhary, 2020*). We thus evaluated the expression levels of ET-1 and ETBR in DRG of adult and aged mice by western blot. For loading controls, we used both ponceau staining for total protein and GAPDH, as we observed lower levels of GAPDH in aged mice. Regardless of the normalization method, we found that ET-1 levels were significantly elevated in aged mice compared to adult (*Figure 4—figure supplement 1A*). ETBR levels were only elevated in aged mice when normalized to the GAPDH control (*Figure 4—figure supplement 1B*). These results confirm that ET-1 levels are elevated in DRG tissue in aged mice.

To determine the role of ETBR signaling on axon growth during aging, DRG explants from aged (21-month-old) mice were treated with BQ788 or vehicle (*Figure 4A*). Aged DRG explants treated

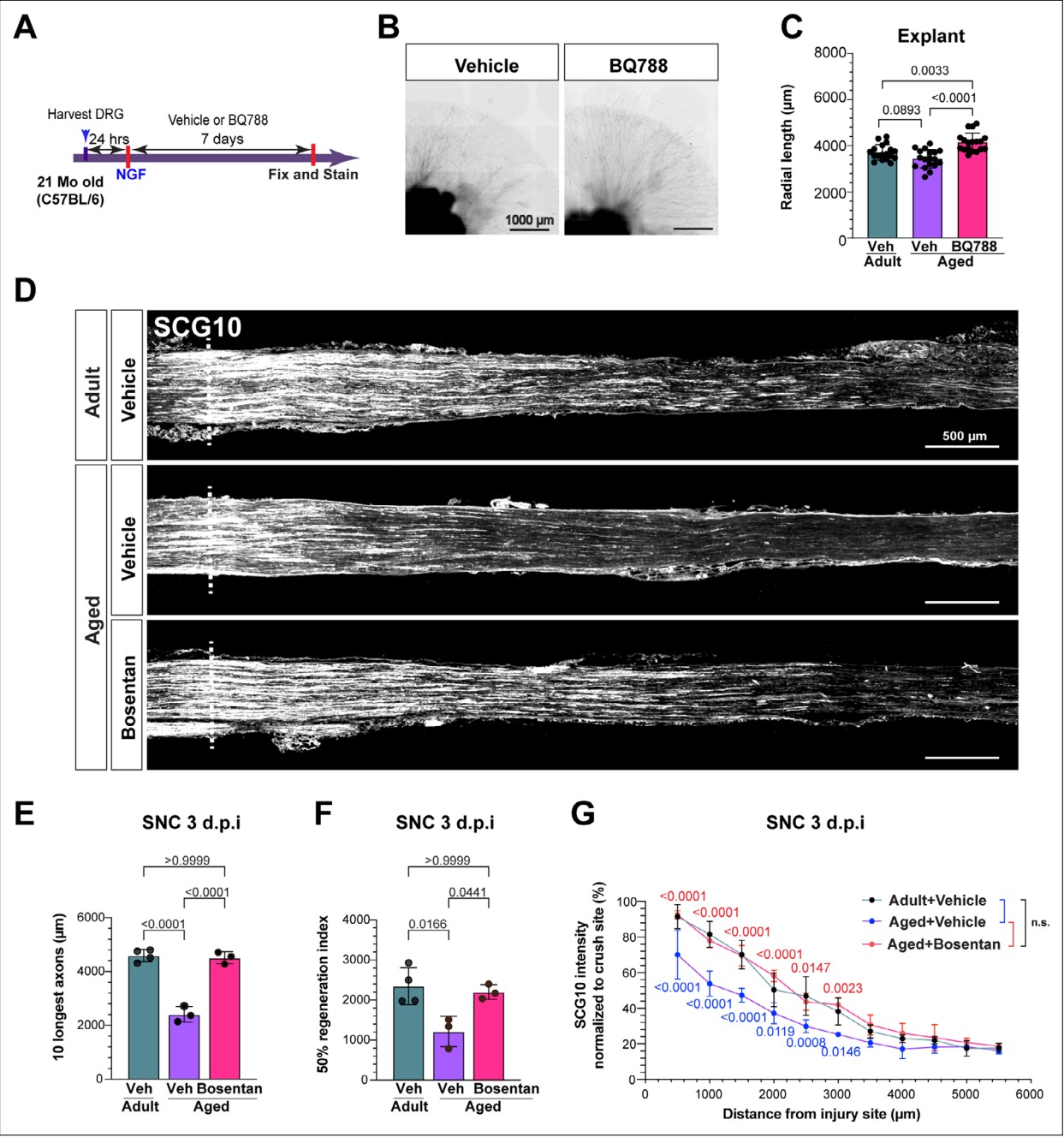

**Figure 4.** Bosentan treatment rescues aging-dependent neuronal regenerative decline. (**A**) Scheme of drug treatment and DRG explant model. (**B**) Representative images of DRG explants 7 days after drug treatment (scale bars, 1000 μm). (**C**) Quantification of radial length of the 50 longest axons from DRG explants. N=36 explants from 6 individual mice (BQ788; 18 replicates, Veh; 18 replicates). (**D**) Representative longitudinal sections of sciatic nerves 3 d after SNC immunostained for SCG10 from mice with the indicated treatment. Dotted line indicates the crush site, determined as the maximal SCG10 intensity (scale bars, 200 μm). (**E, F**) Quantification of the 10 longest axons in indicated groups (**E**). Quantification of 50% regenerative index, calculated as the distance along the nerve where the SCG10 intensity is 50% of the SCG10 intensity at crush site (**F**). (**G**) Quantification of the SCG10 intensity at the indicated distance normalized to the intensity at the crush site for each condition. N (mouse number) = 4(adult, vehicle), 3 (aged, vehicle), and 3 (Bosentan +Age), respectively. The data are presented as mean ± SD.

The online version of this article includes the following source data and figure supplement(s) for figure 4:

**Figure supplement 1.** ET-1 protein expression increases in DRGs of aged mice.

**Figure supplement 1—source data 1.** Original files for western blot analysis.

**Figure supplement 1—source data 2.** PDF file containing original western blots, indicating the relevant bands and treatments.

with BQ788 had significantly increased radial axon length compared to vehicle (*Figure 4B and C*). In fact, aged explants treated with BQ788 had significantly more radial axon growth than that of adult (3-month-old) vehicle explants (*Figure 4C*). In vivo Bosentan pre-treatment in aged mice also increased radial axon growth in DRG explants (*Figure 4—figure supplement 1C–E*). Axon regeneration after peripheral nerve injury in aged mice in vivo was assessed by measuring SCG10 intensity in adult and aged mice 3 days post SNC. As expected, aged vehicle-treated mice had significantly less axon regeneration compared to adult vehicle-treated mice (*Figure 4D–G*; *Zhou et al., 2022*). However, aged mice treated with Bosentan had significantly increased axon regeneration compared to vehicle-treated aged mice. Indeed, treatment with Bosentan increased axon regeneration in aged mice to levels similar to vehicle-treated adult mice (*Figure 4D–G*). These results indicate that inhibition of ETBR rescues the age-dependent decline in axon regeneration.

## Aging affects SGC abundance, transcriptional profile, and morphology

To examine the effect of age on the transcriptional profiles of SGCs, we isolated lumbar DRG from adult and aged mice and performed single nucleus RNA-seq using the Illumina Single Cell 3' RNA Prep (Illumina). We chose this microfluidics-free system for single-nuclei mRNA capture, barcoding, and library prep because it has been shown to be gentler and capture rarer cell populations in the brain (*Frazel et al., 2023*), and our initial attempt with 10 X Genomics on aged tissue yielded insufficient amount of viable cells. After quality control and filtering, we recovered 17,852 high-quality single-nucleus transcriptomes (10,717 adult, 7135 aged) across three biological replicates (one adult, two aged), with an average of 2854 genes and 9899 transcripts detected per cell (*Figure 5—figure supplement 1A and B*). Following RPCA-based integration by age, unsupervised clustering identified 16 distinct cell types (*Figure 5A*). These included four principal neuronal classes (PEP, NF, NP, TH), three glial cell types (SGCs, NMSC, MSC), three previously described fibroblast subsets (EF, EM, PM; *Zhao et al., 2022*) vascular endothelial cells, mural cells, and several immune-related populations (MYEL, TC, BC, HSC; *Figure 5A*). Each cluster was distinguished by multiple unique marker genes, consistent with prior DRG single-cell studies (*Avraham et al., 2020*; *Avraham et al., 2021*; *Jung et al., 2023*; *Mapps et al., 2022*; *Renthal et al., 2020*; *Usoskin et al., 2015*; *Figure 5B*, *Figure 5—source data 1*).

To further investigate aging-associated transcriptional changes in SGCs, we performed differential expression analyses followed by GO (gene ontology; *Figure 5—source data 3*) and KEGG (Kyoto Encyclopedia of Genes and Genomes; *Figure 5—source data 4*) pathway enrichment (*Figure 5—figure supplement 1C and D*). Many genes were upregulated in aged SGCs, including those involved in cell junction assembly, adherens and tight junction pathways, and focal adhesion, indicating significant alterations in cell-cell contacts. In addition, we observed changes in pathways regulating cell projection organization, synapse organization and structure, and supramolecular fiber organization, suggesting widespread remodeling of SGCs architecture. Several signaling pathways implicated in structural and morphogenetic processes, such as axon guidance and Wnt signaling, were also enriched in aged SGCs. Genes related to cellular senescence were also upregulated in aged SGCs (*Figure 5—figure supplement 1D*, *Figure 5—source data 2*). Collectively, these results suggest that aging triggers extensive transcriptional reprogramming in SGCs, reflecting heightened demands for structural integrity, cell junction remodeling, and glia–neuron interactions within the aged DRG microenvironment.

Comparative analysis of relative cell-type abundances between adult and aged DRG revealed a marked decrease in SGCs representation and a corresponding rise in immune cell and neuronal populations in aged samples (*Figure 5C–E*). Changes in morphology of SGCs have been reported with age in rabbit, with a decrease in SGCs number and SGCs retracting and leaving the neuronal soma exposed to the extracellular DRG environment (*Pannese et al., 1997*; *Pannese et al., 1996*). In the DRG of aged mice, neurons appeared to be enveloped by SGCs similarly to adults (*Huang et al., 2006*), but quantification of SGCs number was not performed. We thus examined the impact of age on SGC morphology and number by transmission electron microscopy (TEM). In adult mice, each sensory neuron was enveloped by its own SGCs sheath as described previously (*Pannese, 1981*; *Pannese, 2010*). In aged mice, however, SGCs appeared thinner compared to adult mice (*Figure 5F*). Quantification of the average width of SGCs per neuron demonstrated a significant decrease during aging (*Figure 5G*). Quantification of the number of SGCs nuclei per neuronal soma in TEM images revealed

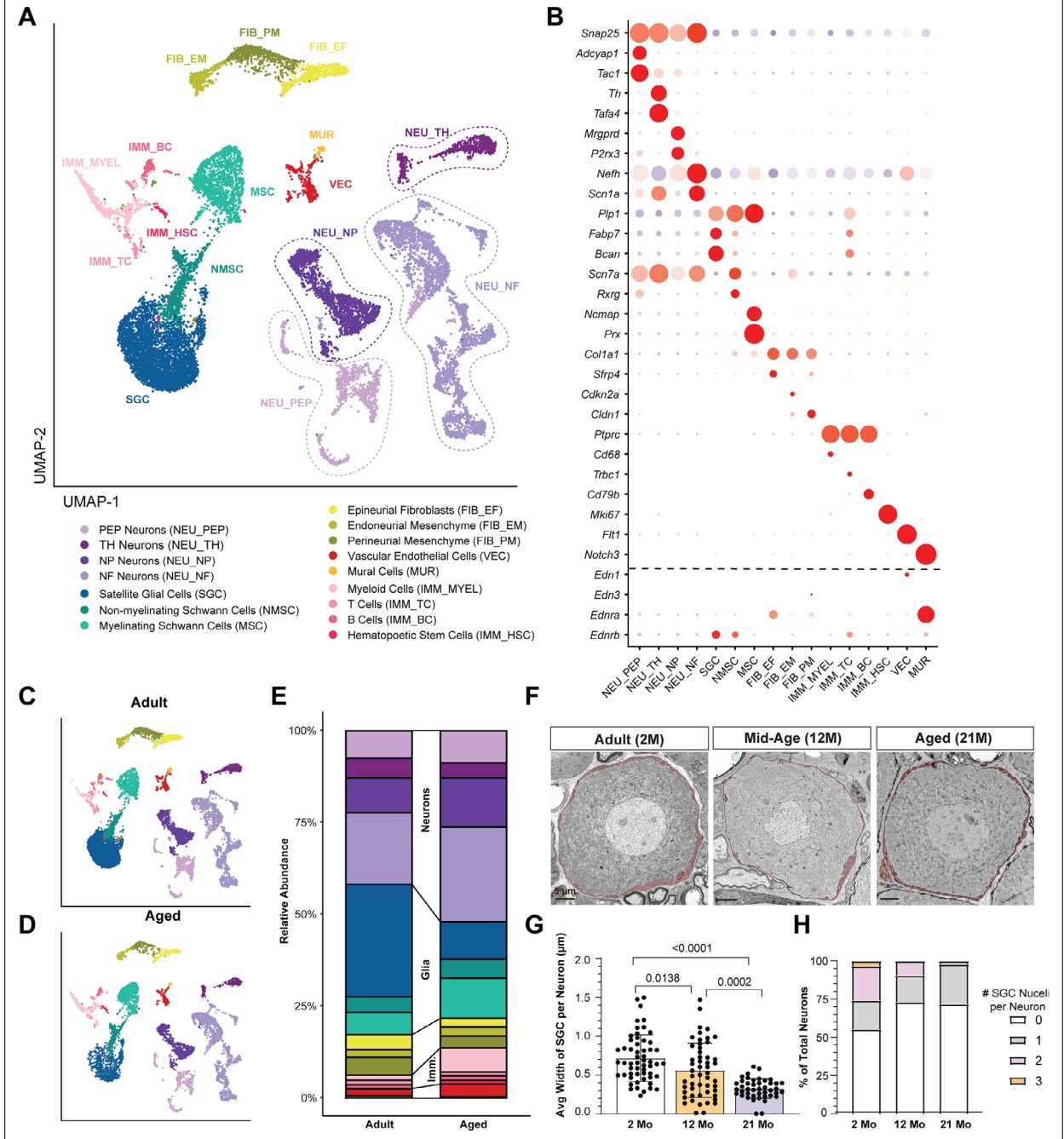

**Figure 5.** Aging alters SGC abundance and morphology. (**A**) UMAP plot of adult and aged snRNA-seq identified 16 cell clusters based on known marker genes. (**B**) Dot plot analysis showing the average gene expression (color coded) and number of expressing cells (dot size) for the marker genes. (**C, D**) UMAP plot of DRG cells from adult (**C**) and aged (**D**) mice. (**E**). Bar plot of cell proportions in DRGs of adult and aged mice. (**F**) Representative TEM images of DRG sections from adult (2 M), middle-aged (12 M), and aged (21 M) mice showing neuronal cell bodies and the enveloping SGCs (SGCs are pseudo-colored in red; scale bars, 5 μm). (**G**) Quantification of the average width of SGC sheath per neuron soma. (**H**) Frequency of neuron soma in TEM images with 0, 1, 2, or 3 SGC nuclei in 2 M, 12 M, and 21 M old mice. The data are presented as mean ± SD.

The online version of this article includes the following source data and figure supplement(s) for figure 5:

**Source data 1.** Marker genes for Illumina snRNA-seq analysis.

**Source data 2.** DGE for SGCs in aged vs adult DRG.

**Source data 3.** GO pathway analysis for SGC in aged vs adult DRG.

**Source data 4.** KEGG pathway analysis for SGC ins aged vs adult DRG.

**Figure supplement 1.** Quality control, DEG, and pathway analysis of aged vs. adult SGCs for Illumina snRNA-Seq.

that 21-month-old mice had fewer SGCs per neuron compared to 2-month-old mice (*Figure 5H*), indicating a decrease in SGCs number with age. These results suggest that, in addition to significant transcriptional changes, SGCs may undergo atrophy during aging.

## ETBR inhibition increases the expression of Cx43 in SGCs in adult and aged mice

Cx43 is a member of the connexin family and can form hemichannels and gap junction channels (*Mazaud et al., 2021*). Several studies suggest that both Cx43 gap junctions and hemichannels operate in SGCs and have an important role in communication between SGCs and between SGCs and sensory neurons (*Hanani and Spray, 2020*; *Retamal et al., 2017*). Cx43 in human SGCs was proposed to play a key role in the protection and maintenance of neurons in spiral ganglia (*Liu et al., 2014*). Past studies in mice have shown that Cx43 expression in SGCs decreases during aging (*Procacci et al., 2008*). In cultured SGCs and astrocytes, endothelin signaling reduces the expression of Cx43 (*Blomstrand et al., 2004*; *Feldman-Goriachnik and Hanani, 2017*; *Rozyczka et al., 2005*). We thus investigated the role of endothelin signaling and aging in Cx43 expression in SGCs. *Gja1*, the gene for Cx43, is enriched in SGCs in both adult and aged mice (*Figure 6A*), as expected (*Procacci et al., 2008*). Changes in *Gja1* expression level with age were statistically difficult to assess because there were fewer SGCs in aged mice compared to adult, and *Gja1* is only detected in 20% of SGCs. We therefore quantified Cx43 protein expression level in SGCs by immunofluorescence (*Figure 6—video 1*). The level of Cx43 expression, measured by both the Cx43 expression area and the number of Cx43 puncta in Fabp7 positive SGCs surrounding neuronal soma, was significantly lower in aged mice compared to adult mice (*Figure 6C and D*), consistent with prior studies (*Ohara et al., 2008*; *Procacci et al., 2008*). Nerve injury increased Cx43 levels in both adult and aged vehicle-treated mice, and Bosentan treatment led to an increase in Cx43 expression in both adult and aged mice (*Figure 6C–E*). Examination of Cx43 expression in SGCs after DRC in adult mice showed no increase in Cx43 levels compared to uninjured mice, but Bosentan-treated mice increased Cx43 levels compared to vehicle (*Figure 6—figure supplement 1A–C*). These results reveal a correlation between Cx43 levels in SGCs and axon regenerative capacity, providing a potential mechanism by which ETBR inhibition with Bosentan may promote axon regeneration after injury (*Figure 6F*). Future experiments will determine if Cx43 acts downstream of ETBR and how Cx43 regulates axon regeneration.

## Discussion

Peripheral nerve injuries have a major impact on patients' functioning and quality of life (*Maita et al., 2023*). The ability of injured neurons to regenerate their axons declines with age (*Geoffroy et al., 2016*; *Geoffroy et al., 2017*; *Pestronk et al., 1980*; *Vaughan, 1992*; *Verdú et al., 2000*), contributing to an increased risk of long-term disability (*Pestronk et al., 1980*; *Vaughan, 1992*; *Verdú et al., 2000*; *Yun, 2015*). Our study shows that ETBR functions in part to limit axonal regenerative capacity and that Bosentan, an FDA-approved ETBR/ETAR antagonist, increases axonal regeneration after peripheral and central axon nerve injury. Furthermore, Bosentan rescues the age-dependent decrease in axonal regenerative capacity. These results suggest that ETBR inhibition may be a beneficial therapeutic to promote axon regeneration after injury.

Our results demonstrate that *Ednrb* is highly enriched in SGCs in the DRG, and selective antagonism of ETBR in DRG mixed cultures and DRG explants increases axon outgrowth. Furthermore, we show that Bosentan treatment in vivo increases axon growth 1 and 3 days after nerve injury, and long-term Bosentan treatment may improve target reinnervation 3 weeks after injury. It is important to note that in the DRG mixed cultures, SGCs, while still present, do not envelop the neuronal soma as they do in the explant culture and in vivo. However, antagonism of ETBR in DRG mixed cultures still increases axon growth. Thus, it is possible that inhibition of ETBR alters the release of trophic factors from SGCs, stimulating the growth potential of neurons and accelerating axon growth. Future studies will be required to unravel how ETBR signaling influences the SGCs secretome and its influence on axon growth. Additionally, orchestration of axon regeneration in the nerve depends on multiple cell types. Polarized vascularization in the nerve precedes Schwann cell migration (*Bhat et al., 2024*; *Cattin et al., 2015*). Since ETBR is expressed in Schwann cells and has been shown to play a role in Schwann cell generation (*Brennan et al., 2000*), we cannot rule out the possibility that Bosentan treatment in

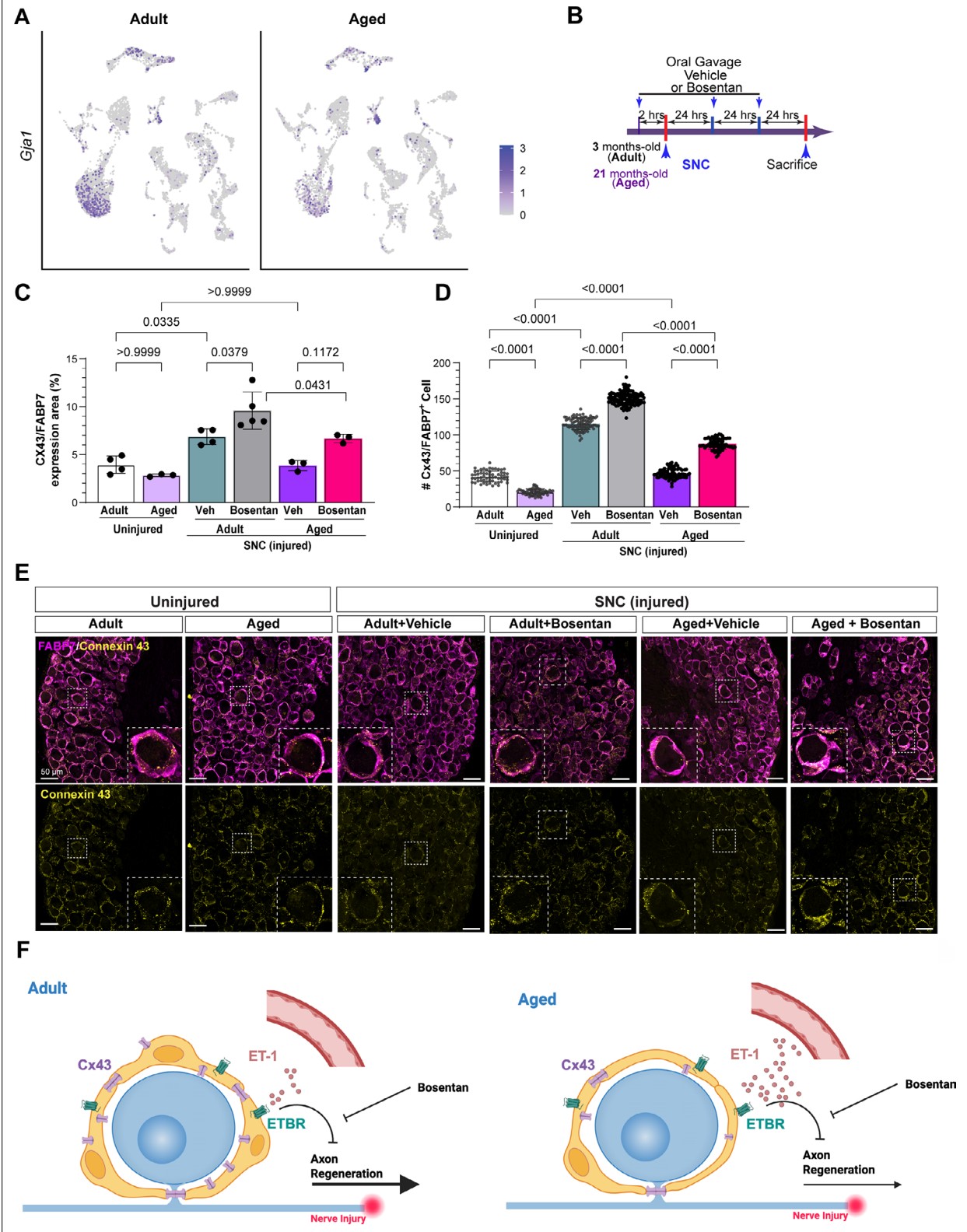

**Figure 6.** ETBR inhibition increases the expression of Cx43 in SGCs in adult and aged mice. (**A**) UMAP overlay for expression of *Gja1* in adult and aged mouse DRG. (**B**) Scheme of drug treatment and peripheral nerve injury model. (**C**) Quantification of the percentage of the Cx43/FABP7 expression area. (**D**) Quantification of the average number of Connexin 43 (Cx43) puncta per FABP7+ cell. The ratio of total Cx43 puncta to the number of FABP7+ cells surrounding a TUJ1+neuron was measured. N(cell number)=60(adult, uninjured), 62(aged, uninjured), 96(vehicle, adult, SNC), 117(bosentan, adult, SNC),

*Figure 6 continued on next page*

*Figure 6 continued*

74(vehicle, aged, SNC), and 74 (bosentan, aged, SNC), respectively. The data are presented as mean ± SD. (**E**) Representative immunostaining images showing Connexin 43 (Cx43), FABP7, and TUJ1 in L4 DRGs from the indicated condition (scale bars, 50 μm). (**F**) Proposed model for the role of ETBR in age-dependent decline in axon regenerative capacity.

The online version of this article includes the following video and figure supplement(s) for figure 6:

**Figure supplement 1.** ETBR inhibition increases the expression of Cx43 in SGCs after DRC.

**Figure 6—video 1.** Z-stack video of DRG section immunostained for FABP7 and CX43.

https://elifesciences.org/articles/100217/figures#fig6video1

vivo may affect axon regeneration through ETBR in Schwann cells in addition to SGCs. However, given our results demonstrating that DRG explants maintain the SGCs-sensory neuron morphology, and axon outgrowth in this model does not rely on repair Schwann cells, we can conclude that blocking ETBR signaling in SGCs contributes to increasing axon outgrowth, at least in the initial axon growth phase. Together, these results suggest a novel molecular mechanism in peripheral nerve regeneration.

We hypothesize that the major ligand acting on ETBR in the DRG is ET-1. This is based on our scRNA-seq data in the DRG showing *Edn1*/ET-1 enrichment in endothelial cells, as well as the dense vascularization in DRG. Additionally, the observation that elevated tissue or plasma concentrations of ET-1 occurs with age (*Jankowich and Choudhary, 2020*) and our results showing that ET-1 levels increase in the aged DRG support the notion that ETBR signaling in SGCs limits axon regenerative capacity in aged mice. However, other studies have suggested that ET-1 is also expressed by small diameter neurons in human DRG (*Giaid et al., 1989*), thus it is possible that neuronally derived ET-1 may act on ETBR in SGCs. Additionally, our scRNA-seq data shows that another ligand for ETBR, *Edn3*/ET-3, is enriched in fibroblasts in the DRG. While ET-1 binds to ETAR and ETBR with the same affinity, ET-3 shows a higher affinity to ETBR than to ETAR (*Davenport et al., 2016*). It also has been shown in cultured cortical astrocytes that application of either ET-1 or ET-3 leads to a robust inhibition of connexin Cx43 expression (*Rozyczka et al., 2005*). Thus, we cannot exclude the possibility that ET-3 may act on ETBR in SGCs to regulate axon growth. Lastly, as our scRNA-seq data shows that *Ednra* is enriched in mural cells (*Chen et al., 2000*), it is also possible that endothelin's impact on vascular permeability through ETAR activity contributes to axon regeneration. Future experiments will be required to test how injury and age affect vascular permeability in the DRG. Nonetheless, whether via ET-1 or ET-3, our results highlight that ETBR acts to limit spontaneous sensory axon regeneration, and that inhibition of ETBR promotes axon regeneration.

Although a biologically intrinsic mechanism to inhibit axon regeneration appears counterintuitive, other mechanisms limiting axon regenerative capacity have been reported. Axonally synthesized proteins typically support nerve regeneration through retrograde signaling and local growth mechanisms (*Sahoo et al., 2018*). RNA binding proteins (RBP) are needed for this process and other aspects of post-transcriptional regulation of neuronal mRNAs. One such RBP, the RNA binding protein KHSRP, is locally translated following nerve injury, promotes decay of other axonal mRNAs, and slows axon regeneration (*Patel et al., 2022*). Additionally, the Rho signaling pathway is an inhibitory regulatory mechanism that slows the growth of spiral ganglion neurite in culture (*Lie et al., 2010*). These mechanisms may function to limit plasticity and prevent maladaptive neural rewiring that can happen after injury (*Costigan et al., 2009*), but can also hinder beneficial recovery after injury. Whether ETBR inhibition over a longer time period has consequences on target innervation and functional recovery will require further investigation.

Investigation of the effects of aging on sensory ganglia has shown that the number of neurons decreases with age in different species including humans, cats, and rats (*Aldskogius and Risling, 1989*; *Nagashima and Oota, 1974*; *Schmalbruch, 1987*). The effect of age on SGCs has received less attention, but we know that the number of SGCs decreases with age in the rabbit (*Pannese et al., 1997*; *Pannese et al., 1996*). In agreement with these prior studies, our snRNA-seq and morphology data reveal a marked decrease in SGCs representation in the DRG of aged mice. Our EM data indicate that the number of SGCs surrounding each neuronal soma decreases with age and appears thinner, suggesting SGCs atrophy in addition to SGCs loss. KEGG pathway analysis also suggests that aged SGCs transition into a senescent phenotype during aging, a state where cells no longer divide but remain viable. SGCs have the capacity to divide, although this process is very slow (*Maniglier et al.,*

*2022*). It was proposed that SGCs self-renewal maintain the SGCs population through adulthood, ensuring neuronal survival throughout life (*Maniglier et al., 2022*). This balance may become dysfunctional with age, leading to the accumulation of senescent SGCs. In the nerve, senescence of Schwann cells in aged mice was shown to undermine axonal regeneration, and systemic elimination of senescent cells with senolytic drugs improves axonal regeneration (*Fuentes-Flores et al., 2023*), raising the interesting possibility that senescent SGCs also contribute to limit axon regeneration in aged mice. Future studies are needed to determine if aged SGCs, similarly to aged Schwann cells, secrete inhibitory factors that limit axon regeneration. We also observed changes in pathways regulating cell projection organization, synapse organization and structure, and supramolecular fiber organization, suggesting widespread remodeling of SGCs architecture. Several signaling pathways implicated in structural and morphogenetic processes, such as axon guidance and Wnt signaling, were also enriched in aged SGCs. Collectively, these results suggest that aging triggers extensive transcriptional reprogramming in SGCs, reflecting heightened demands for structural integrity, cell junction remodeling, and glia–neuron interactions within the aged DRG microenvironment.

Cx43 is a transmembrane protein that performs canonical gap junction functions, such as connecting adjacent cells and allowing the diffusion of ions and small molecules. Cx43 can also form hemichannels that may have an important role in communication between SGCs and sensory neurons (*Retamal et al., 2017*). Emerging evidence suggests that connexins can perform non-channel functions, such as protein interaction, cell adhesion, and intracellular signaling (*Mazaud et al., 2021*). Using our snRNA-seq data to compare *Gja1* levels in adult vs aged SGCs was statistically difficult to interpret, as there were significantly fewer SGCs in aged mice. The histology examining Cx43 protein levels was normalized to total SGCs and showed a significant decrease in Cx43 puncta/SGC in aged mice, consistent with previous studies and the hypothesis that a prominent decrease in Cx43 is a marker of senescence (*Procacci et al., 2008*). It has also been shown in SGCs of aged mice that there is an increase in gap junctions and dye coupling (*Huang et al., 2006*). This apparent discrepancy could be explained by expression changes in connexin types other than Cx43 that may contribute to the gap junctions. Indeed, our snRNA-seq data revealed changes in other connexins with age, but the percentage of cells expressing these genes and the level of expression was not robust enough to draw meaningful conclusions. Deeper sequencing would likely be required to verify this hypothesis. It is also important to note that, while sequencing is a strong tool, aging has profound effects on post-translational regulation, such as altered protein half-life, defects in protein trafficking, and increased protein degradation (*Santos and Lindner, 2017*; *Skariah and Todd, 2021*). Thus, in conjunction with sequencing, examination of protein levels of various connexins in SGCs will be necessary.

Our results show that Cx43 in SGCs from both adult and aged injured mice increased with Bosentan treatment compared to vehicle treatment. Studies in cultured SGCs and astrocytes have demonstrated that endothelin signaling reduces gap junction coupling (*Blomstrand et al., 2004*; *Feldman-Goriachnik and Hanani, 2017*; *Rozyczka et al., 2005*). Thus, we hypothesize that ETBR inhibition in SGCs contributes to axonal regeneration by increasing Cx43 levels, gap junction coupling, or hemichannels, facilitating SGCs-neuron communication. However, it is also possible that other mechanisms downstream of ETBR signaling in SGCs contribute to axon regeneration. For example, glutamate uptake by SGCs is regulated by glutamate transporters GLT-1 and GLAST and is crucial for neuronal function (*Berger and Hediger, 2000*; *Hanani and Verkhratsky, 2021*; *Ohara et al., 2008*). Studies in astrocytes suggest that endothelin exposure reduces glutamate uptake by decreasing transporter expression. One of the functions of GLAST in astrocytes is related to lesion-induced plasticity in the developing brain (*Takasaki et al., 2008*), suggesting that ETBR might limit axon regeneration by influencing SGCs' glutamate reuptake. As ETBR is a G-protein-coupled receptor which interacts with Gi and Gq (but not Gs; *Ham et al., 2024*; *Tani et al., 2024*), its inhibition could have numerous effects on intracellular signaling pathways and transcriptional response in SGCs. Further studies are needed to determine the detailed molecular mechanisms by which ETBR signaling in SGCs limits axon regeneration.

Taken together, our results suggest that ETBR signaling limits axon regeneration after nerve injury and contributes to age-related decreases in neuronal regenerative capacity. Bosentan, an FDA-approved ETBR/ETAR antagonist, significantly increased axon regeneration in both adult and aged mice, suggesting a potential therapeutic avenue that may be used to enhance axonal regeneration after nerve injury.

# Materials and methods

**Key resources table**

| Reagent type (species) or resource | Designation | Source or reference | Identifiers | Additional information |
|---|---|---|---|---|
| Strain, strain background (*Mus musculus*) | C57BL/6 | Envigo; Jackson Laboratory | Envigo: 027; JAX: 000664 RRID:IMSR_CRL:027 RRID:IMSR_JAX:000664 | Female and male, used at various ages |
| Genetic reagent (*M. musculus*) | Ai14 | Jackson Laboratory | JAX: 007914 RRID:IMSR_JAX:007914 | B6.Cg-Gt(ROSA)26Sortm14(CAG-tdTomato)Hze/J |
| Genetic reagent (*M. musculus*) | Fabp7CreER | Toshihiko Hosoya (gift) | | Crossed with Ai14 to generate Fabp7CreER::Ai14 |
| Biological sample (*M. musculus*) | DRG tissue | This paper | | L3-L5, L4-L5 DRGs isolated from adult and aged mice |
| Chemical compound, drug | Bosentan | Sigma-Aldrich | Sigma: PHR2708 | 10 mg/kg, oral gavage |
| Chemical compound, drug | Ambrisentan | Tocris | Tocris: 5828 | 10 mg/kg, oral gavage |
| Chemical compound, drug | BQ788 | Sigma-Aldrich | B157 | 1 µM for in vitro use |
| Chemical compound, drug | BQ123 | R&D Systems | 1188 | 1 mM for in vitro use |
| Chemical compound, drug | IRL620 | Sigma-Aldrich | SCP0135 | 100 nM for in vitro use |
| Antibody | Rabbit anti-ETBR (polyclonal) | Abcam | ab117529 RRID:AB_10902070 | WB (1:500) |
| Antibody | Mouse anti-ET-1 (monoclonal) | Invitrogen | MA3-005 RRID:AB_2096246 | WB (1:500) |
| Antibody | Rabbit anti-GAPDH (polyclonal) | Cell Signaling | 5174 s RRID:AB_10622025 | WB (1:5000) |
| Antibody | Rabbit anti-FABP7 (polyclonal) | Invitrogen | PA5-24949 RRID:AB_2542449 | IHC (1:1000) |
| Antibody | Rabbit anti-STMN2 (SCG10) (polyclonal) | Novus/Techne | NBP1-49461 RRID:AB_10011569 | IHC (1:1000) |
| Antibody | Rabbit anti-Cx43 (polyclonal) | Cell Signaling | 3512s RRID:AB_2294590 | IHC (1:200) |
| Antibody | Mouse anti-TUJ1 (βIII tubulin) (monoclonal) | Biolegend | 801202 RRID:AB_2313773 | IHC (1:1000) |
| Antibody | Rabbit anti-PGP9.5 (polyclonal) | LS Bio | LS-B5981-50 | IHC (1:500) |
| Antibody | Secondary antibodies Alexa Fluor 488/594/647 | Invitrogen | Various | IHC (1:500) |
| Commercial assay, kit | RNAscope Fluorescent Multiplex Kit | ACD (Advanced Cell Diagnostics) | | For RNA in situ hybridization |
| Commercial assay, kit | RNeasy Mini Kit | QIAGEN | 74104 | RNA extraction |
| Commercial assay, kit | High-Capacity cDNA Reverse Transcription Kit | Thermo Fisher | 4368814 | cDNA synthesis |
| Commercial assay, kit | PowerUp SYBR Green Master Mix | Thermo Fisher | A25780 | qPCR |
| Commercial assay, kit | LIVE/DEAD Fixable Aqua Dead Cell Stain Kit | Thermo Fisher | L34965 | Cell viability stain |
| Chemical compound | DAPI | Sigma-Aldrich | D9542 | (300 nM) nuclear stain |
| Commercial assay, kit | ProLong Gold Antifade Mountant | Invitrogen | P36930 | For mounting fluorescent samples |
| Chemical compound | PFA (paraformaldehyde) | Various | | 4% used for fixation |
| Chemical compound | OCT compound | Tissue-Tek | | For cryosectioning |

*Continued on next page*

*Continued*

| Reagent type (species) or resource | Designation | Source or reference | Identifiers | Additional information |
|---|---|---|---|---|
| Peptide, recombinant protein | NGF (Nerve Growth Factor) | Alomone | N-240 | Used in DRG explants |
| Chemical compound | HBSS | Thermo Fisher, Gibco | 14175–079 | Dissection medium |
| Chemical compound | HEPES | Thermo Fisher, Gibco | 15630080 | Buffering agent |
| chemical compound | Papain | Worthington Biochemical | LS003126 | For tissue dissociation |
| Chemical compound | L-cysteine | Sigma | C7352 | Added to dissociation mix |
| Chemical compound | DNase I | Worthington Biochemical | LS002139 | For DNA degradation during dissociation |
| Chemical compound | Collagenase | Sigma | C6885 | Enzyme for tissue dissociation |
| Chemical compound | Neurobasal-A Medium | Thermo Fisher, Gibco | 12349015 | Culture medium |
| Commercial assay, kit | B-27 Plus Supplement | Thermo Fisher, Gibco | A3582801 | Culture supplement |
| Commercial assay, kit | GlutaMAX Supplement | Thermo Fisher, Gibco | 35050061 | Glutamine substitute |
| Chemical compound | Poly-D-lysine | | | Coating coverslips |
| Biological sample (*M. musculus*) | DRG explants | This paper | | For explant culture |
| Software, algorithm | Fiji | *Schindelin et al., 2012* | RRID:SCR_002285 | Image analysis |
| Software, algorithm | QuantStudio 6 Flex System | Thermo Fisher | | For qPCR analysis |
| Software, algorithm | Seurat v5.1.0 | Satija Lab | RRID:SCR_007322 | For sc/snRNA-seq analysis |
| Software, algorithm | CellRanger v7.1.0 | 10 X Genomics | RRID:SCR_017344 | For scRNA-seq processing |
| Software, algorithm | Pipseeker v3.3.0 | Fluent BioSciences | | For snRNA-seq processing |
| Software, algorithm | Imaris v9.7 | Oxford Instruments | RRID:SCR_007370 | Vessel quantification |

## Study design

The main goals of this study were to investigate the role of ETBR in axonal regeneration, explore its impact on SGCs in the DRG, and evaluate the therapeutic potential of Bosentan, an FDA-approved drug, on axon regeneration and age-related decline capacity for nerve repair. We performed scRNA-seq and snRNAseq analysis on DRG samples from adult and aged mice. We analyzed samples from mice subjected to various treatments and performed in vitro assays including immunofluorescence (IF), western blotting, quantitative real-time polymerase chain reaction (qPCR), RNA in situ hybridization (RNAscope), and TEM. For in vitro cell assays, data were collected from at least three independent cultures, as indicated in figure legends. In vivo and ex vivo experiments were conducted using biological replicates, as denoted by the 'n' values in the figure legends. The sample sizes were determined based on previous experience for each experiment, and mice were randomly assigned to the experimental groups whenever possible. No mice, outliers, or other data points were excluded. Details on animal assignment, randomization, and blinding in different experiments are found in the corresponding sections describing each experiment in Materials and methods.

## Animals

Mice of different age groups were included in the study, specifically 2–3 month-old (adult, female and male), 12-month-old (mid-age, female), and 21-month-old (aged, female). Wild-type C57BL/6 mice were purchased from Envigo (Envigo #027) and Jackson Laboratory (Stock No: 000664). Ai14 (B6. Cg-Gt(ROSA)26Sortm14(CAG-tdTomato)Hze/J, JAX Stock No: 007914) mice were obtained from The Jackson Laboratory (*Madisen et al., 2010*). The *Blbp^CreER* (*Fabp7^CreER*) mouse line was a generous gift from Dr. Toshihiko Hosoya (*Maruoka et al., 2011*). Ai14 mice were crossed with *Fabp7^CreER* mice to obtain *Fabp7^CreER*::Ai14 mice. Mice were housed in the animal facility at Washington University in St. Louis, where temperature (64–79 °F) and humidity (30%–70%) were carefully controlled. They were socially housed in individually ventilated cages, with 1–5 mice per cage, and subjected to a 12 hr light/dark cycle (6 am/6 pm). Mice had unrestricted access to food and water throughout the study. All experimental procedures were conducted following the approved protocol (21–0104) by the

Institutional Animal Care and Use Committees of Washington University in St. Louis. All experiments adhered to relevant guidelines and regulations. The study obtained approval from the Washington University School of Medicine Institutional Animal Care and Use Committee (IACUC) under protocol A-3381–01. The mice were housed and cared for in the animal care facility at Washington University School of Medicine, which is accredited by the Association for Assessment & Accreditation of Laboratory Animal Care (AALAC) and complies with the PHS guidelines for Animal Care. The facility has been accredited since 7/18/97, and its USDA Accreditation Registration number is 43 R-008.

## Primary adult DRG culture

L4 and L5 DRGs were collected from 12-week-old mice and placed in cold dissection medium composed of HBSS (Thermo Fisher, Gibco; Catalog#: 14175–079) with 10% 1 M HEPES (Thermo Fisher, Gibco; Catalog#: 15630080). The DRGs were then transferred to freshly prepared pre-warmed dissociation medium containing 15 U/mL Papain suspension (Worthington Biochemical; Catalog#: LS003126), 0.3 mg/mL L-cysteine (Sigma; Catalog#: C7352), 0.1 mg/mL Deoxyribonuclease I (Worthington Biochemical; Catalog#: LS002139), and 10% 1 M HEPES in HBSS. The samples were incubated at 37°C for 20 min. After washing the samples twice with pre-warmed HBSS, collagenase (150 µg/mL; Sigma; Catalog#: C6885) was added, and the samples were incubated at 37°C for another 20 min. Following two additional washes with pre-warmed HBSS, the resulting single-cell suspension was gently triturated and resuspended in complete medium consisting of Neurobasal-A Medium (Thermo Fisher, Gibco; Catalog#: 12349015) supplemented with B-27 Plus Supplement (Thermo Fisher, Gibco; Catalog#: A3582801) and GlutaMAX Supplement (Thermo Fisher, Gibco; Catalog#: 35050061). The cell suspension was then passed through 70 µm cell strainers. The single-cell suspension was then centrifuged at 500×rpm at 4°C for 5 min, and the cell pellet was resuspended in complete Neurobasal medium. Cells were seeded on poly-D-lysine (PDL)-coated 18 mm coverslips at a density of $1.0 \times 10^3$ cells per coverslip. For drug treatment, BQ788 (Sigma; Catalog #B157) 1 µM, BQ123 (R&D system; Catalog #: 1188) 1 mM, IRL620 (Sigma; Catalog #: SCP0135) 100 nM or DMSO as vehicle control were added directly to the complete medium just before suspending the cells and seeding them into separate culture plates and incubated for the entire 24 hr culture period. The animals were randomized into groups, and the researcher was not blinded during the analysis.

## DRG explant culture

L3-L5 DRGs from male and female 8-week- or 21-month-old mice were collected into cold explant media [Putrescine Dihydrochloride (Sigma Aldrich, Catalog# p7505), L-Glutamine (Thermofisher, Catalog# A2916801), Insulin Transferrin Selenium (Sigma Aldrich, Catalog# A4034), Glucose (Sigma Aldrich, Catalog# G7021), Basal Medium Eagle (Thermofisher, Catalog# 35050061), Bovine Serum Albumin (Sigma Aldrich, Catalog# A9430), Ham's F-12 Nutrient Media (Thermo Fisher, Catalog# 11765–054)]; DRG roots were trimmed to 2 mm with a sharp scalpel, then placed into a 15 µl drop of a 1:1 ratio of explant media and ECM Gel from Engelbreth-Holm-Swarm murine sarcoma (Sigma Aldrich, Catalog# E1270) in cell culture plates. 24 hr after plating, Nerve Growth Factor (NGF; Alomone, Catalog# N-240) was added to the media. Note that NGF does not promote significant effects on axon growth in DRG explants, but stimulates glial cell migration (*Klimovich et al., 2020*). We opted to include NGF in our explant assay to increase the potential of stimulating axon regeneration with pharmacological manipulations of ETBR. Seven days after plating, explants were fixed and stained for TUJ1 to quantify axon length.

## Sciatic nerve and dorsal root injuries

Sciatic nerve crush injuries were performed following established protocols (*Avraham et al., 2021*). Briefly, 12-week-old mice were anesthetized using 1.5% inhaled isoflurane. A small skin incision was made to expose the sciatic nerve at mid-thigh level, approximately 1.2 cm from the L4 DRG. The sciatic nerve was fully crushed for 10 s using 0.1 mm forceps (#55). The wound was closed with wound clips, and the mice were placed on a warming pad until fully awake. At the designated time points post-surgery, L4 and L5 DRG were dissected for further analysis. The animals were randomized into groups.

DRC injuries were performed as previously described (*Avraham et al., 2021*). Surgery was conducted on 12-week-old mice under 1.5% inhaled isoflurane anesthesia. A small midline skin

incision (~1 cm) was made over the thoracic vertebrae at L2-L3, followed by paraspinal muscle release and stabilization of the vertebral column with metal clamps under the L2-L3 transverse processes. Dorsal laminectomy was performed using forceps at the L2-L3 level, and the right L3-L5 dorsal roots were crushed simultaneously for 5 s. The proximity between L3-L5 roots resulted in a crush distance of 1–2 mm to L3 DRG, 4–5 mm to L4 DRG, and 7–8 mm to L5 DRG. The crushing process exerted force on the roots, causing disruption of nerve fibers without interrupting the endoneurial tube. Paraspinal muscles were sutured using 6–0 sutures (Ethicon, Catalog#: J212H), and the wound was closed with clips. Mice were placed on a warming pad until fully awake, and L4 DRG and L4 dorsal roots were dissected at 3 days post-injury. The animals were randomized into groups.

### Drug administration

To assess the in vivo functions of endothelin receptors, pharmacological testing was conducted. For the analysis of nerve regeneration at one day post-injury, oral gavage administration of Ambrisentan (Tocris, Catalog# 5828, 10 mg/kg body weight; *Kappes et al., 2020*) and Bosentan (Sigma, Catalog#PHR2708, 10 mg/kg body weight; *Pinho-Ribeiro et al., 2014*) was performed 2 hr before the injury, with sample collection taking place 24 hr after the injury. For the analysis of nerve regeneration after 3 days post-injury, Ambrisentan (10 mg/kg body weight) and Bosentan (10 mg/kg body weight) were administered via oral gavage 2 hr before the injury and once daily after the injury. The treatments were randomized within the surgery mice group.

### Western blotting

L4-L5 DRGs were collected and lysed in RIPA Buffer (Cell Signaling, catalog #9806) and heated at 99°C for 10 min. The protein lysates were then loaded onto a 10% Ready Gel Tris-HCl Precast Gels (Bio-Rad, catalog#1611119) in running Buffer (Bio-Rad, Catalog#1610744) and transferred to 0.2 μm PVDF membranes (Bio-Rad, Catalog#1620216). The membranes were blocked in 5% non-fat dry milk (Bio-Rad, Catalog#1706404XTU) Tris-Buffered Saline (TBS) containing 0.1% Tween-20 at pH 7.6 for 1 hr. Following blocking, the membranes were incubated overnight at 4°C with primary antibodies Rabbit-anti-ETBR (Abcam, Catalog# ab117529, 1:500) Mouse-anti-ET-1 1:500 (Invitrogen, Catalog# MA3-005) Rabbit-anti-GAPDH 1:5000 (Cell Signaling, Catalog# 5174 s) in blocking buffer. Afterward, the membranes were washed three times with TBST (Tris-buffered saline with 0.1% Tween-20) and then incubated with horseradish peroxidase-conjugated anti-rabbit (Invitrogen, Catalog# 31460, 1:5000) or anti-mouse (Invitrogen, Catalog # 31430, 1:5000) antibodies in blocking buffer for 1 hr at room temperature. Subsequently, the membranes were washed three times with TBST and developed using SuperSignal West Dura (Thermo Fisher Scientific, Catalog# 34075). The ChemiDoc System (Bio-Rad) was used to image the membranes. For subsequent analysis, the membranes were stripped using a mild stripping buffer composed of (1.5% glycine, 0.1% SDS, and 1% Tween –20, pH 2.2 in ddH$_2$O). Following stripping, the protocol was repeated for GAPDH staining (Santa Cruz Biotechnology, Catalog #sc-51907, 1:8000) as the loading control. The resulting images were analyzed using Fiji software. To quantify the protein levels, the intensity of ET-1 or ETBR was normalized to the intensity of GAPDH. The researcher was not blinded during the analysis.

### Quantitative real-time PCR

For qRT-PCR, L4 and L5 DRGs of each mouse were collected and total RNA was extracted using RNeasy Mini Kit (QIAGEN, Cat# 74104). For cDNA synthesis, 500 ng of RNA was converted into cDNA with the High-Capacity cDNA Reverse Transcription Kit (Thermo Fisher, Catalog# 4368814) according to manufacturer's specifications. Quantitative PCR was completed using the PowerUp SYBR Green Master Mix (Thermo Fisher, Cat# A25780) using gene-specific primers (resource table) from Primerbank (https://pga.mgh.harvard.edu/primerbank/). qRT-PCR was performed on a QuantStudio 6 Flex System. Expression fold change for each gene of interest was calculated using the ΔCq method and normalized to the expression fold change of *Gadph* expression compared to controls. The researcher was not blinded during the analysis. The detail for each primer is listed in .

### Immunohistochemistry

Mice were euthanized with CO$_2$ asphyxiation and transcardially perfused with PBS followed by 4% paraformaldehyde (PFA). PFA-fixed tissues were incubated in 30% sucrose in phosphate-buffered

saline (PBS) overnight at 4°C, specimens were embedded in optimal cutting temperature compound (OCT) (Tissue-Tek), stored at −80°C until further processing. Transverse sections of DRG (L4) and longitudinal sections of sciatic nerve were cut on a cryostat at 10 μm and stored at –20°C until processed. Before staining, sections were warmed to room temperature and dried on a 60°C slide warmer for 5 min. Sections were treated with a blocking solution containing 4% normal donkey serum (NDS; LAMPIRE; Catalog#7332100) with 0.5% Triton X-100 in PBS for 1 hr at room temperature. Then samples were incubated in the primary antibodies, which were diluted in 1% NDS with 0.3% Triton X-100 in PBS overnight at 4°C. After three PBS rinses, samples were incubated with Alexa Fluor–conjugated secondary antibodies in PBS with 0.3% Triton X-100 in 1 hr, followed by incubation in 300 nM 4',6-diamidino-2-phenylindole (DAPI, Sigma-Aldrich, Catalog# D9542) at room temperature for 10 min. Samples were rinsed before mounting with ProLong Gold Antifade Mountant (Invitrogen, Catalog#:P36930). For co-staining of Cx43 and Fabp7, both of which are rabbit antibodies, the tissue sections were first blocked with 10% NDS in PBS containing 0.3% Triton X-100 at room temperature for 1 h. Subsequently, the sections were incubated overnight at 4°C with rabbit anti-Cx43 antibody diluted in 0.1% Triton X-100 in PBS. After three rinses with PBS, the samples were incubated with an excess of conjugated Fab Fragment secondary antibody in PBS with 0.3% Triton X-100 for 1 hr. Following three washes with PBS, the sections were incubated with rabbit anti-Fabp7 antibody, diluted in 0.1% Triton X-100 in PBS, at room temperature for 2 hr. After three PBS washes, the samples were incubated with Alexa Fluor-conjugated secondary antibodies in PBS with 0.3% Triton X-100 for 1 hr. Finally, the sections were incubated with DAPI at room temperature for 10 minutes, followed by rinsing and mounting with ProLong Gold Antifade Mountant. DRG sections were imaged with a confocal laser-scanning microscope (Zeiss LSM880). Figures showing large longitudinal sciatic nerve sections were produced using EVOS M7000 Imaging System with image stitching and/or stack software. For adult DRG culture staining, neurons were fixed in pre-warmed 1% PFA in 7.5% sucrose at 37°C for 15 min. Then post-fixed in prewarmed 2% PFA in 15% sucrose for 30 min. Rise with PBS three times, the fixed neurons were directly for immunostaining following the same staining protocol described above. Cultured neurons were imaged with acquired using the ECLIPSE Ti2 inverted microscope.

Whole-mount DRG immunostaining was performed by injecting 0.1 μL of Lycopersicon Esculentum Lectin (Vector, Catalog# DL-1178–1) through the tail vein to label the blood vessels (*Robertson et al., 2015*). L4 DRGs were dissected after 20 mins and fixed overnight in 4% PFA at 4°C. Following a previously established protocol (*Yang et al., 2017*), DRGs underwent a series of washing and permeabilization steps with 0.3% Triton X-100 in PBS, repeated every hour for 5 hr. Subsequently, the tissues were incubated with primary antibodies in a blocking solution consisting of 75–0.3% PBST, 20% DMSO, and 5% Donkey Serum for 4 days, followed by further washing with 0.3% Triton X-100 in PBS every hour for 5 hr. The tissues were then incubated with secondary antibodies in the blocking solution for 2 days, with subsequent washing using 0.3% Triton X-100 in PBS every hour for 5 hrs. All of these washing and incubation procedures were carried out on a rocking platform at room temperature. The tissues were dehydrated in methanol for 1 hr and cleared using a 1:2 mixture of Benzyl Alcohol to Benzyl Benzoate. Imaging was carried out using an LSM880 confocal microscope, and blood vessel reconstruction was performed with Imaris 9.7.

Primary antibodies included rabbit anti-FABP7 (Invitrogen, Catalog#:PA5-24949, 1:1000); rabbit anti-STMN2(SCG10) (Novus a bio-techne, Catalog#:NBP1-49461, 1:1000); rabbit anti-Connexin-43 (Cell Signaling, Catalog#: 3512 s, 1:200); mouse anti TUJ1 (β-III tubulin) (Biolegend, Catalog#:801202, 1:1000). Secondary antibodies conjugated to Alexa Fluor 488, Alexa Fluor 594, and Alexa Fluor 647 (Invitrogen), conjugated Fab Fragment Donkey anti Rabbit Alexa Fluor (Jackson ImmunoResearch Labs) were diluted 1:500.

For intraepidermal nerve fiber density measurements, fixed, free-floating 50-μm-thick sections of footpads from the hind limbs were stained with rabbit anti-Protein Gene Product 9.5 (1:500, lSBIO, Cat# LS-B5981-50), followed by AF594 conjugated secondary antibody (1:500, Invitrogen, Cat# A-21207). Sections were mounted and coverslipped using VECTASHIELD anti-fade mounting media with DAPI (Vector Laboratory, Cat# H-2000). PGP 9.5 positive intraepidermal nerve fibers (IENFs) crossing into the epidermis were counted from Z-stack images using the 20 x objective on a Nikon W1 CSU SoRa. IENF densities were averaged from three to four sections for each animal. Counting was performed with the observer blinded to the treatment group.

## RNAscope in situ hybridization

The RNAscope fluorescent multiplex reagent kit (Advanced Cell Diagnostics, ACD) was used according to the manufacturer's instructions. Slides were retrieved from a –80°C freezer, washed with PBS to remove OCT, and dried on a 60°C slide warmer for 5 min. Post-fixation was performed by immersing the slides in cold (4°C) 4% paraformaldehyde (PFA) for 15 min. Tissues were dehydrated using a series of ethanol washes (50% ethanol for 5 min, 70% ethanol for 5 min, and 100% ethanol for 10 min) at room temperature. Slides were air-dried briefly and then incubated in RNAscope Target Retrieval buffer (ACD; Catalog# 322000) at 98–100°C for 5 min. After rinsing in 100% ethanol, hydrophobic boundaries were drawn around each section using a hydrophobic pen (ImmEdge PAP pen; Vector Labs). Once the boundaries were dry, protease III reagent was added to each section and incubated for 15 min. Slides were briefly washed in PBS at room temperature. Each slide was placed in a prewarmed humidity control tray (ACD) with dampened filter paper, and a mixture of probes was pipetted onto each section until fully submerged. The slides were then incubated in a HybEZ oven (ACD) at 40°C for 2 hr. Following the probe incubation, the slides were washed twice with 1 x RNAscope wash buffer and returned to the oven for 30 min after submersion in AMP-1 reagent. This washing and amplification process was repeated using AMP-2, AMP-3, and AMP-4 reagents with incubation periods of 15 min, 30 min, and 15 min, respectively. For all mouse experiments, AMP-4 ALT A (Channel 1=Atto 488, Channel 2=Alexa 550, Channel 3=Atto 647) was used. Slides were then washed twice with 0.1 M phosphate buffer (PB, pH 7.4). Subsequently, the slides were processed with 300 nM DAPI at room temperature for 10 min and cover-slipped using Prolong Gold Antifade mounting medium. DRG sections were imaged using a confocal laser-scanning microscope (Zeiss LSM880).

## Transmission electron microscopy

For transmission electron microscopy (TEM), mice were perfused PBS and then with the fixative (2.5% glutaraldehyde and 4% paraformaldehyde in 0.1 M Cacodylate buffer), followed by post-fixation in same fixation 24 hr at 4 C. For secondary post fixation, the samples were rinsed in 0.15 M cacodylate buffer containing 2 mM calcium chloride 3 times for 10 min each followed by a secondary fixation in 1% osmium tetroxide and 1.5% potassium ferrocyanide in 0.15 M cacodylate buffer containing 2 mM calcium chloride for 1 hr in the dark. The samples were then rinsed 3 times for 10 min each in ultrapure water and en bloc stained with 2% aqueous uranyl acetate overnight at 4°C in the dark. After 4 washes for 10 min each in ultrapure water, the samples were dehydrated in a graded acetone series (10%, 30%, 50%, 70%, 90%, 100% x3) for 10 min each step, infiltrated with Spurr's resin (Electron Microscopy Sciences), and embedded and polymerized at 60°C for 72 hr. The samples from adult mice were processed as above except they were dehydrated in a graded ethanol series (10%, 30%, 50%, 70%, 90%, 100% x3) for 10 min each step, infiltrated with Epon resin (Electron Microscopy Sciences), and embedded and polymerized at 60°C for 48 hr. Post-curing, 70-nm-thin sections were cut, post-stained with 2% aqueous uranyl acetate and Sato's lead, and imaged on a TEM (Jeol JEM-1400 Plus) at 120 kV.

## Single cell RNA sequencing

L4 and L5 DRGs from mice were collected and placed into cold dissection medium consisting of Hank's Balanced Salt Solution (HBSS; ThermoFisher, Gibco; Catalog#: 14175–079) supplemented with 10% 1 M HEPES (Thermo Fisher, Gibco; Catalog#: 15630080). Subsequently, the ganglia were transferred to freshly prepared pre-warmed dissociation medium composed of 15 U/mL Papain suspension (Worthington Biochemical; Catalog#: LS003126), 0.3 mg/mL L-cysteine (Sigma; Catalog#: C7352), 0.1 mg/mL Deoxyribonuclease I (Worthington Biochemical; Catalog#: LS002139), and 10% 1 M HEPES in HBSS. The samples were then incubated at 37°C for 20 min. After two washes with pre-warmed HBSS, the ganglia were incubated with collagenase (150 µg/mL; Sigma; Catalog#: C6885) at 37°C for 20 min. Following an additional two washes with pre-warmed HBSS, the resulting single cell suspension was resuspended by gently triturating in complete medium consisting of Neurobasal-A Medium (Thermo Fisher, Gibco; Catalog#: 12349015) supplemented with B-27 Plus Supplement (Thermo Fisher, Gibco; Catalog#: A3582801) and GlutaMAX Supplement (Thermo Fisher, Gibco; Catalog#: 35050061). The cell suspension was then passed through 70 µm cell strainers. The single-cell suspension was further resuspended in HBSS containing 10% 1 M HEPES and 0.1% Fetal Bovine Serum (FBS; Thermo Fisher; Catalog#: A3160401). The cells were stained with the LIVE/DEAD Fixable Aqua Dead Cell Stain Kit (Thermo Fisher; Catalog#: L34965) to identify live cells. Live single cells were sorted

using the MoFlo cell sorter (Beckman Coulter, Indianapolis, IN). The sorted cells were washed with a PBS +0.04% BSA solution and manually counted using a hemocytometer. The cell solution was adjusted to a concentration of 700–1000 cells/µL and loaded onto the 10 X Chromium system.

Single-cell RNA-Seq libraries were prepared using the Chromium Next GEM Single Cell 3′ (v3.1 Kit) from 10 X Genomics. Read alignment and transcript counting were performed with the CellRanger (v7.1.0) package from 10 X Genomics and 19,356 cells with >300 detected genes were retained for downstream analysis. Sample QC, principal component analysis, clustering, DGE, quantification, and statistical analysis were performed using the Seurat package (v5.1.0). Following normalization with SCTransform, principal component analysis, and graph-based clustering, four low-quality and/or doublet clusters were identified and removed based on the following characteristics: statistical enrichment (DGE p<0.05) of mixed major cell type marker genes or mitochondria-specific transcripts, and/or low average genes per cell (<1000). After removal of low-quality clusters, cells with disproportionately high mitochondrial transcript levels (calculated as the fraction of reads mapping to mitochondrial genes) or UMI counts were filtered by setting a cutoff at the median plus five median absolute deviations (MAD) for each metric. The remaining 16,764 high-quality cells were visualized using UMAP (Uniform Manifold Approximation and Projection). Differential gene expression (DGE) analyses were performed in the Seurat package with a Wilcoxon Rank Sum test, and genes with Bonferroni-adjusted p-values <0.05 were considered for downstream analysis and visualization.

## Single nucleus RNA sequencing

L3, L4, and L5 DRGs were collected from mice and immediately placed on dry ice. DRG were stored in –80 until processing. Nuclei were isolated using the Illumina Nuclei Isolation Kit (Illumina; Catalog# 20132795). The DRG were dissociated using a KIMBLE 2 mL Dounce homogenizer (Sigma; Catalog# D8938) in Nuclei Extraction Buffer (NEB) and passed through a 40 µm cell strainer. The nuclei were briefly centrifuged and then passed through a 10 µm cell strainer. The nuclei were resuspended in Nuclei Suspension Buffer (NSB) and counted for intact nuclei using 7-AAD stain (Sigma; Catalog# SML1633). Nuclei were processed using the Illumina Single Cell 3′ RNA Prep, T20 Kit (Illumina; Catalog# FB0005366, FB0005373, FB0005362, FB0005382).

Single-nuclei RNA-Seq libraries were prepared using the Illumina V Library Prep Kit (Illumina; Catalog# 20132789, FB0005374, FB0005373). Read alignment and transcript counting were performed with the Pipseeker (v3.3.0) package from Fluent BioSciences and 19,270 cells with >1000 genes were retained for downstream analysis (11,521 adult, 7749 aged). Sample QC, integration, clustering, DGE, quantification, and statistical analysis were performed using the Seurat package (v5.1.0). Following normalization with SCTransform, RPCA batch integration, and graph-based clustering, five low-quality and/or doublet clusters were identified and removed based on the following characteristics: statistical enrichment (DGE p<0.05) of mixed major cell type marker genes or mitochondria-specific transcripts, and/or low total cell count (<50). The remaining 17,852 high-quality cells were visualized using UMAP. Differential gene expression (DGE) analyses were performed in the Seurat package with a Wilcoxon Rank Sum test, and genes with Bonferroni-adjusted p-values <0.05 were considered for downstream analysis and visualization. KEGG pathway and GO enrichment were performed using the clusterProfiler package in R. The researcher was not blinded during analysis.

## Images acquisition and quantification

Confocal images were captured under the LSM880 confocal microscope (Carl Zeiss) unless otherwise specified. The images were captured under the following parameters: 1 × optical zoom, scan speed 6, averaged 2 times, a pinhole of 1 AU, 1024 × 1024 pixel size. The z-stack images were projected into overlay images using the 'Maximum Intensity Projection' function in Zen Black software (Carl Zeiss). Sciatic nerve longitudinal images, the spinal cord and brain images were captured under the EVOS M7000 Imaging System (Invitrogen). The images were captured under 10x objective in 1980 × 1080 pixel size. The stitched nerve images generated by the EVOS imaging system. The neuronal culture images were captured under the ECLIPSE Ti2 inverted microscope (Nikon). The images were captured under 10x objective in 2304 × 2304 pixel size. Three tissue sections were analyzed on average for each independent mouse. The researcher performing this analysis was not blinded during analysis.

For quantifying Cx43 expression area in the DRG, the CX43$^+$ and Fabp7$^+$ area was measured using the 'Analysis Particles' function in Fiji. To quantify CX43, confocal imaging with Z-stacks (10 µm) was

performed using an LSM880 microscope. The acquired images were subsequently used for 3D reconstruction using Imaris 9.7 software. The staining area of FABP7 underwent initial surface reconstruction, while the CX43 puncta were isolated through spot reconstruction at a resolution of 0.1 μm x 0.2 μm. The average number of CX43 puncta was determined for each FABP7$^+$ cell (*Figure 6—video 1*). For quantifying the regeneration index in the injured sciatic nerve section, the highest SCG10 intensity along the nerve was defined as the crush site, as described (*Cho et al., 2013*; *Feng et al., 2023*). The average SCG10 intensity at distances away from the crush site was normalized to the SCG10 intensity at the crush site using the 'measure' function in Fiji. For quantifying the neuron regenerative capacity in cultured DRG neurons, TUJ1-stained neurons were segmented by thresholding and subjected to neurite tracing using 'Simple Neurite Tracer' plug-in in Fiji. The researcher was not blinded during analysis.

DRG explant analysis was performed by creation of an ROI around each explant. The ROI was removed for organoid contrast and the radial length was measured from the outer peripheral section of the ROI. The 35 longest axons in each image were measured (μm) and averaged.

For TEM, SGC width around the neuron was measured by averaging 4 individual points that did not contain an SGC nucleus around the neuron using Fiji/ImageJ software. Frequency of neurons with 0, 1, 2, or 3 SGC nuclei per neuron was quantified by counting each image by eye.

## Statistical analysis

All statistical analyses and graphs were conducted, organized, and generated in GraphPad Prism 9. Numerical data were presented as mean ± SD from at least three independent animals. Group mean difference was analyzed by either two-tailed unpaired Student's t-test or one-way ANOVA with Bonferroni post-hoc test. For experiments including groups and multiple time-point measurements, data were analyzed by two-way ANOVA with Bonferroni post-hoc test. p-Values below 0.05 were considered as significant difference. The number of animals in each group is presented in figure legends. No a priori exclusion criteria or statistical power calculations were used.

## Acknowledgements

We thank members of the Cavalli lab and the Mokalled lab for valuable discussions and suggestions. We gratefully acknowledge Michael Savio from The Alvin J Siteman Cancer Center at Barnes-Jewish Hospital and Washington University School of Medicine for assistance with single cell sorting. We also acknowledge the assistance of John Wulf II, Gregory Strout and Dr. Sanja Sviben at the Washington University Center for Cellular Imaging (WUCCI) in electron microscopy studies, which is supported by Washington University School of Medicine, The Children's Discovery Institute of Washington University and St. Louis Children's Hospital (CDI-CORE-2015–505 and CDI-CORE-2019–813) and the Foundation for Barnes-Jewish Hospital (3770 and 4642). TEM images were acquired using an AMT Nanosprint15-MkII sCMOS camera, which was purchased with support from the Office of Research Infrastructure Programs (ORIP), a part of the NIH Office of the Director under grant OD032186. The authors show their gratitude and respect to all animals sacrificed in this study.

## Additional information

### Competing interests

Michael B Thomsen: is affiliated with CS27 Bioinformatics. The other authors declare that no competing interests exist.

### Funding

| Funder | Grant reference number | Author |
| --- | --- | --- |
| National Institute of Neurological Disorders and Stroke | NS122260 | Valeria Cavalli |

| Funder | Grant reference number | Author |
| --- | --- | --- |
| National Institute of Neurological Disorders and Stroke | NS111719 | Valeria Cavalli |
| National Institute of Neurological Disorders and Stroke | NS115492 | Valeria Cavalli |

The funders had no role in study design, data collection and interpretation, or the decision to submit the work for publication.

## Author contributions

Rui Feng, Sarah F Rosen, Conceptualization, Data curation, Formal analysis, Investigation, Methodology, Validation, Visualization, Writing – original draft, Writing – review and editing; Irshad Ansari, Data curation, Formal analysis, Methodology, Validation, Visualization, Writing – review and editing; Sebastian John, Formal analysis, Methodology, Validation, Writing – review and editing; Michael B Thomsen, Data curation, Formal analysis, Investigation, Methodology, Validation, Visualization, Writing – review and editing; Oshri Avraham, Methodology, Validation, Visualization, Writing – review and editing; Cedric G Geoffroy, Conceptualization, Investigation, Resources, Writing – review and editing; Valeria Cavalli, Conceptualization, Data curation, Funding acquisition, Investigation, Project administration, Supervision, Writing – original draft, Writing – review and editing

## Author ORCIDs

Rui Feng ⓘ https://orcid.org/0000-0001-5724-9550
Sarah F Rosen ⓘ https://orcid.org/0000-0001-8844-1203
Valeria Cavalli ⓘ https://orcid.org/0000-0001-9978-050X

## Ethics

All experimental procedures were conducted following the approved protocol (21-0104) by the Institutional Animal Care and Use Committees of Washington University in St. Louis. All experiments adhered to relevant guidelines and regulations. The study obtained approval from the Washington University School of Medicine Institutional Animal Care and Use Committee (IACUC) under protocol A-3381-01. The mice were housed and cared for in the animal care facility at Washington University School of Medicine, which is accredited by the Association for Assessment & Accreditation of Laboratory Animal Care (AALAC) and complies with the PHS guidelines for Animal Care. The facility has been accredited since 7/18/97, and its USDA Accreditation Registration number is 43-R-008.

Reviewer #1 (Public review): https://doi.org/10.7554/eLife.100217.3.sa1
Reviewer #2 (Public review): https://doi.org/10.7554/eLife.100217.3.sa2
Author response https://doi.org/10.7554/eLife.100217.3.sa3

# Additional files

## Supplementary files

Supplementary file 1. List of qPCR Primers.

MDAR checklist

## Data availability

All data associated with this study are presented in the paper or the supplementary materials. All western blot raw images are included as source files where indicated. scRNA and snRNA sequencing data have been deposited at NCBI Gene Expression Omnibus (GEO) under accession number GSE298413 and GSE298412. Further information about data presented is available from the corresponding author upon reasonable request.

The following datasets were generated:

| Author(s) | Year | Dataset title | Dataset URL | Database and Identifier |
|---|---|---|---|---|
| Feng R, Rosen SF, Ansari I, John S, Thomsen MB, Avraham O, Geoffroy CG, Cavalli V | 2025 | Endothelin B receptor inhibition rescues aging-dependent neuronal regenerative decline [snRNA-seq] | https://www.ncbi.nlm.nih.gov/geo/query/acc.cgi?acc=GSE298413 | NCBI Gene Expression Omnibus, GSE298413 |
| Feng R, Rosen SF, Ansari I, John S, Thomsen MB, Avraham O, Geoffroy CG, Cavalli V | 2025 | Endothelin B receptor inhibition rescues aging-dependent neuronal regenerative decline [scRNA-seq] | https://www.ncbi.nlm.nih.gov/geo/query/acc.cgi?acc=GSE298412 | NCBI Gene Expression Omnibus, GSE298412 |

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
