## [Editor Report · eLife Assessment]

This **important** study examines the role of endothelin signaling in nerve regeneration, providing **convincing** evidence that it functions as a default brake on axon regrowth. Inhibiting endothelin signaling with Bosentan promotes regeneration and counteracts the decline in regenerative potential caused by aging. Since Bosentan is an FDA-approved drug, these findings could have therapeutic value in clinical settings where peripheral nerve regeneration is not adequate or seriously impaired, as is often the case in older individuals.

---

## [Referee Report · Reviewer #1 (Public review)]

The manuscript by Feng et al. reported that Endothelin B receptor (ETBR) expressed by the satellite glial cells (SGCs) in the dorsal root ganglions (DRG) acted to inhibit sensory axon regeneration in both adult and aged mice. Thus, pharmacological inhibition of ETBR with specific inhibitors resulted in enhanced sensory axon regeneration in vitro and in vivo. In addition, sensory axon regeneration significantly reduces in aged mice and inhibition of ETBR could restore such defect in aged mice. Moreover, the study provided some evidence that the reduced level of gap junction protein connexin 43 might act downstream of ETBR to suppress axon regeneration in aged mice. Overall, the study revealed an interesting SGC-derived signal in the DRG microenvironment to regulate sensory axon regeneration. It provided additional evidence that non-neuronal cell types in the microenvironment function to regulate axon regeneration via cell-cell interaction.

However, the molecular mechanisms by which ETBR regulates axon regeneration are unclear, and the structure of the manuscript is relatively not well organized, especially the last section. Some discussion and explanation about the data interpretation are needed to improve the manuscript.

(1) The result showed that the level of ETBR was not changed after the peripheral nerve injury. Does it mean that its endogenous function is to limit the spontaneous sensory axon regeneration? In other words, the results suggest that SGCs expressing ETBR or vascular endothelial cells expressing its ligand ET-1 act to suppress sensory axon regeneration. Some explanation or discussion about this are necessary. Moreover, does the protein level of ETBR or its ligand change during aging?

(2) In ex vivo experiments, NGF was added in the culture medium. Previous studies have shown that adult sensory neurons could initiate fast axon growth in response to NGF within 24 hours. In addition, dissociated sensory neurons could also initiate spontaneous regenerative axon growth without NGF after 48 hours. Some discussion or rationale is needed to explain the difference between NGF-induced or spontaneous axon growth of culture adult sensory neurons and the roles of ETBR and SGCs.

(3) In cultured dissociated sensory neurons, inhibiting ETBR also enhanced axon growth, which meant the presence of SGCs surrounding the sensory neurons. Some direct evidence is needed to show the cellular relationship between them in culture.

(4) In Figure 3, the in vivo regeneration experiments first showed enhanced axon regeneration either at 1 day or 3 days after the nerve injury. The study then showed that inhibiting ETBR could enhance sensory axon growth in vitro from uninjured naïve neurons or conditioning lesioned neurons. To my knowledge, in vivo sensory axon regeneration is relatively slow during the first 2 days after the nerve injury and then enter the fast regeneration mode in the 3rd day, representing the conditioning lesion effect in vivo. Some discussion is needed to compare the in vitro and the in vivo model of axon regeneration.

(5) In Figure 5, the study showed that the level of connexin 43 increased after ETBR inhibition in either adult or aged mice, proposing an important role of connexin 43 in mediating the enhancing effect of ETBR inhibition on axon regeneration. However, in the study there was no direct evidence supporting that ETBR directly regulate connexin 43 expression in SGCs. Moreover, there was no functional evidence that connexin 43 acted downstream of ETBR to regulate axon regeneration.

In the revised manuscript, most comments have been addressed with some new experiments or text revisions in the results or discussion. For representative images showing in vitro cultured DRG neurons, it would be much more convincing if several neurons in the same imaging field are shown, rather than a single neuron (Figure 2A, 3J).

---

## [Referee Report · Reviewer #2 (Public review)]

Summary:

Feng and colleagues set out to investigate the effect of manipulating endothelin signaling on nerve regeneration, focusing on the crosstalk between endothelial cells (ECs) in dorsal root ganglia (DRG), which secrete ET-1, and satellite glial cells (SGCs), which express the ETBR receptor. ETBR signaling limits axon growth. Using in vitro explant assays coupled with pharmacological inhibition in mouse models of nerve injury, the authors demonstrate that the ETAR/ETBR antagonist Bosentan promotes axon regeneration, and that this effect is maintained in aged mice. Although Bosentan inhibits both endothelin receptors A and B, comparison with an ETAR-specific antagonist suggests primary involvement of the ET-1/ETBR pathway. In the DRG, ETBR is mostly expressed by SGCs, a cell type implicated in nerve regeneration. SGCs ensheath and couple with DRG neurons through gap junctions formed by Cx43. The pro-regenerative effects of ETBR inhibition are attributed in part to an increase in Cx43 levels, which are expected to enhance neuron-SGC coupling. snRNA sequencing and TEM analysis reveal a decline in SGC numbers, morphological changes, and transcriptional reprogramming that may impair their pro-regenerative capacity.

Strengths:

The study is well-executed, and the main conclusion (that ETBR signaling inhibits axon regeneration after nerve injury and contributes to the age-related decline in regenerative capacity) is well supported by the data. In addition, the study highlights the importance of vascular signals in nerve regeneration, a topic that has gained traction in recent years. Importantly, these results further emphasize the contribution of long-neglected SGCs to nerve tissue homeostasis and repair. Although the study does not provide a complete mechanistic understanding, the findings are robust and are likely to attract the interest of a broad readership.

Weaknesses:

While certain aspects could have been further addressed experimentally, these points were either technically challenging or considered beyond the scope of the current study, and are appropriately addressed in the Discussion.

(1) It remains to be determined whether the accelerated axon regrowth observed after nerve injury depends on cellular crosstalk mediated by ET-1 at the lesion site. Are ECs along the nerve secreting ET-1? What cells are present in the nerve stroma that could respond and participate in the repair process? Would these interactions be sensitive to Bosentan? Dissecting these contributions would require cell-specific manipulations. The potential roles of ECs, fibroblast and SCs in the nerve are discussed.

(2) It is suggested that the permeability of DRG vessels may facilitate the release of vascular-derived signals. The possibility that the ET-1/ETBR pathway modulates vascular permeability, and that this in turn contributes to the observed effects on regeneration, is discussed.

(3) It cannot be excluded that ET-3 in fibroblasts is relevant for controlling SGC responses. The possibility that both ET-1 and ET-3 participate in ETBR- dependent effect on axon regeneration is discussed.

(4) The discovery that ET-1/ETBR signaling in SGC curtails the growth capacity of axons at baseline raises questions about the physiological role of this pathway. This remains to be elucidated with cell type-specific knockout approaches.

(5) The modulation of Cx43 expression by ET-1/ETBR is examined by immunostaining, but a complementary analysis by quantitative RT-PCR on sorted SGCs would have been a valuable addition. However, quantifying Cx43 on purified SGCs was not attainable due to technical complications.

(6) The conclusion "that ETBR inhibition in SGCs contributes to axonal regeneration by increasing Cx43 levels, gap junction coupling or hemichannels and facilitating SGC-neuron communication" are consistent with previous studies (Procacci et al., 2008) but in apparent discrepancy with increased gap junctions and dye coupling in SGCs of aged mice (Huang et al., 2006). More experiments are required to clarify what distinguishes a beneficial increase in coupling after ETBR inhibition, from what is observed in aging.

(7) The effect of Bosentan likely extends beyond the modulation of Cx43 levels. Cell type-specific knockout of Cx43 and ETBR, studies of SGCs-neuron coupling, and biochemical analysis of Cx43 functions would clarify the link between ETBR, Cx43 regulation, and axon regeneration. A discussion of alternative mechanisms is provided.

---

## [Author Response]

The following is the authors’ response to the original reviews.

**Reviewer #1 (Public Review):**
The manuscript by Feng et al. reported that the Endothelin B receptor (ETBR) expressed by the satellite glial cells (SGCs) in the dorsal root ganglions (DRG) acted to inhibit sensory axon regeneration in both adult and aged mice. Thus, pharmacological inhibition of ETBR with specific inhibitors resulted in enhanced sensory axon regeneration in vitro and in vivo. In addition, sensory axon regeneration significantly reduces in aged mice and inhibition of ETBR could restore such defect in aged mice. Moreover, the study provided some evidence that the reduced level of gap junction protein connexin 43 might act downstream of ETBR to suppress axon regeneration in aged mice. Overall, the study revealed an interesting SGC-derived signal in the DRG microenvironment to regulate sensory axon regeneration. It provided additional evidence that non-neuronal cell types in the microenvironment function to regulate axon regeneration via cell-cell interaction.However, the molecular mechanisms by which ETBR regulates axon regeneration are unclear, and the manuscript's structure is not well organized, especially in the last section. Some discussion and explanation about the data interpretation are needed to improve the manuscript.

We thank the reviewer for the positive comments. We agree that the mechanisms by which ETBR signaling functions as a brake on axon growth and regeneration remain to be elucidated. We believe that unraveling the detailed molecular pathways downstream of ETBR signaling in SGCs that promote axon regeneration is beyond the scope of this manuscript. Answering these questions would first require cell specific KO of ETBR and Cx43 to confirm that this pathway is operating in SGCs to control axon regeneration. We would also need to identify how SGCs communicate with neurons to regulate axon regeneration, which is a large area of ongoing research that remains poorly understood. Our data showing that pharmacological inhibition of ETBR with specific FDA-approved inhibitors enhances sensory axon regeneration provide not only new evidence for non-neuronal mechanisms in nerve repair, but also a new potential clinical avenue for therapeutic intervention.

As suggested by the reviewer, we have extensively revised the organization of the manuscript, especially the last section of results. We have performed additional snRNAseq experiments to establish the impact of aging in DRG. We have also performed additional experiments to determine if blocking ETBR improves target tissue reinnervation. Following the reviewer’s suggestion, we have also expanded the Discussion section to discuss alternative mechanisms and offer additional interpretation of our data. Below we describe how we address each point in detail.

(1) The result showed that the level of ETBR did not change after the peripheral nerve injury. Does this mean that its endogenous function is to limit spontaneous sensory axon regeneration? In other words, the results suggest that SGCs expressing ETBR or vascular endothelial cells expressing its ligand ET-1 act to suppress sensory axon regeneration. Some explanation or discussion about this is necessary. Moreover, does the protein level of ETBR or its ligand change during aging?

We thank the reviewer for this point. Our results indeed indicate that one endogenous function of ETBR is to limit the extent of sensory axon regeneration. This may be a part of a mechanism to limit spontaneous sensory axon growth or plasticity and maladaptive neural rewiring after nerve injury. While the increased growth capacity of damaged peripheral axons can lead to reconnection with their targets and functional recovery, the increased growth capacity can also lead to axonal sprouting of the central axon terminals of injured neurons in the spinal cord, and to pain (see for example Costigan et al 2010, PMID: 19400724). In the context of aging that we describe here, this protective mechanism may hinder beneficial recovery. Other mechanisms that slow axon regeneration have been reported, and include, for example, axonally synthesized proteins, which typically support nerve regeneration through retrograde signaling and local growth mechanisms. RNA binding proteins (RBP) are needed for this process. One such RBP, the RNA binding protein KHSRP is locally translated following nerve injury. Rather than promoting axon regeneration, KHSRP promotes decay of other axonal mRNAs and slows axon regeneration. Another example includes the Rho signaling pathway, which was shown to function as an inhibitory mechanism that slows the growth of spiral ganglion neurites in culture. We have now included these examples in the Discussion section.

To address the reviewer’s second question, we have checked protein levels of ETBR and ET-1 in adult and aged DRG tissue. We observed a robust increase in ET-1 in aged DRG, while the levels of ETBR did not appear to change significantly. These results are now presented in Figure 4- Figure Supplement 1, and further support the notion that in aging, activation of the ETBR signaling hinders axon regeneration.

(2) In ex vivo experiments, NGF was added to the culture medium. Previous studies have shown that adult sensory neurons could initiate fast axon growth in response to NGF within 24 hours. In addition, dissociated sensory neurons could also initiate spontaneous regenerative axon growth without NGF after 48 hours. Some discussion or rationale is needed to explain the difference between NGF-induced or spontaneous axon growth of culture adult sensory neurons and the roles of ETBR and SGCs.

We appreciate the reviewer’s suggestion. In adult DRG explant or dissociated cultures, NGF is not typically required for survival or axon outgrowth. However, in dissociated culture, the addition of NGF to the medium stimulates growth from more neurons compared to controls (Smith and Skene 1997). In the DRG explant, NGF does not promote significant effects on axon growth, but stimulates glial cell migration (Klimovich et al 2020). We opted to included NGF in our explant assay to increase the potential of stimulating axon regeneration with pharmacological manipulations of ETBR. We have now clarified these considerations in the Method section.

(3) In cultured dissociated sensory neurons, inhibiting ETBR also enhanced axon growth, which meant the presence of SGCs surrounding the sensory neurons. Some direct evidence is needed to show the cellular relationship between them in culture.

We thank the reviewer for raising this point and have added new data, now presented in Figure 2B, to show that in mixed DRG cultures, SGCs labeled with Fabp7 are present in the culture in proximity to neurons labeled with TUJ1, but they do not fully wrap the neuronal soma. These results are consistent with prior findings reporting that as time in culture progresses, SGCs lose their adhesive contacts with neuronal soma and adhere to the coverslip (PMID: 22032231, PMID: 27606776). While in some cases SGCs can maintain their association with neuronal soma in the first day in culture after plating, in our hands, most SGCs have left the soma at the 24h time point we examined.

(4) In Figure 3, the in vivo regeneration experiments first showed enhanced axon regeneration either 1 day or 3 days after the nerve injury. The study then showed that inhibiting ETBR could enhance sensory axon growth in vitro from uninjured naïve neurons or conditioning lesioned neurons. To my knowledge, in vivo sensory axon regeneration is relatively slow during the first 2 days after the nerve injury and then enters the fast regeneration mode on the 3rd day, representing the conditioning lesion effect in vivo. Some discussion is needed to compare the in vitro and the in vivo model of axon regeneration.

We agree that axon growth is relatively slow the first 2 days and enters a fast growth mode on day 3. This has been elegantly demonstrated in Shin et al Neuron 2012 (PMID: 22726832), where an in vivo conditioning injury 3 days prior increases axon growth one day after injury. In vitro, similar effects have been described: a prior in vivo injury accelerates growth capacity within the first day in culture, but a similar growth mode occurs in naive adult neurons after 2-3 days in vitro (Smith and Skene 1996). We also know that the neurite growth in culture is stimulated by higher cell density, likely because non-neuronal cells can secrete trophic factors (Smith and Skene 1996). Our in vitro results thus suggest that blocking ETBR in SGCs in these mixed cultures may alter the media towards a more growth promoting state. In vivo, our data show that Bosentan treatment for 3 days partially mimics the conditioning injury and potentiate the effect of the conditioning injury. One possible interpretation is that inhibition of ETBR alters the release of trophic factors from SGCs. Future studies will be required to unravel how ETBR signaling influence the SGCs secretome and its influence on axon growth. We have now included these discussions points in the Results and Discussion Section.

(5) In Figure 5, the study showed that the level of connexin 43 increased after ETBR inhibition in either adult or aged mice, proposing an important role of connexin 43 in mediating the enhancing effect of ETBR inhibition on axon regeneration. However, in the study, there was no direct evidence supporting that ETBR directly regulates connexin 43 expression in SGCs. Moreover, there was no functional evidence that connexin 43 acted downstream of ETBR to regulate axon regeneration.

We thank the reviewer for this point and agree that we do not provide direct evidence that connexin 43 acts downstream of ETBR to regulate axon regeneration. To obtain such functional evidence would require selective KO of ETBR and Cx43 in SGCs, which we believe is beyond the scope of the current study. We have revised the Results and Discussion sections to emphasize that while we observe that ETBR inhibition increases Cx43 levels and Cx43 levels correlates with axon regeneration, whether Cx43 directly mediates the effect on axon regeneration remains to be established. We also discuss potential alternative mechanisms downstream of ETBR in SGCs that could contribute to the observed effects on axon regeneration. Specifically, we discuss the possibility that ETBR signaling may limit axon regeneration via regulating SGCs glutamate reuptake functions, because of the following reasons: (1) Similarly to astrocytes, glutamate uptake by SGCs is important to regulate neuronal function, (2) exposure of cultured cortical astrocytes to endothelin results in a decrease in glutamate uptake that correlates with a major loss of basal glutamate transporter expression (GLT-1 and1), (3) Both glutamate transporters are expressed in SGCs in sensory ganglia (4) GLAST and glutamate reuptake function is important for lesion-induced plasticity in the developing somatosensory cortex.

**Reviewer #2 (Public Review):**
Summary:In this interesting and original study, Feng and colleagues set out to address the effect of manipulating endothelin signaling on nerve regeneration, focusing on the crosstalk between endothelial cells (ECs) in dorsal root ganglia (DRG), which secrete ET-1 and satellite glial cells (SGCs) expressing ETBR receptor. The main finding is that ETBR signaling is a default brake on axon growth, and inhibiting this pathway promotes axon regeneration after nerve injury and counters the decline in regenerative capacity that occurs during aging. ET-1 and ETBR are mapped in ECs and SGCs, respectively, using scRNA-seq of DRGs from adult or aged mice. Although their expression does not change upon injury, it is modulated during aging, with a reported increase in plasma levels of ET-1 (a potent vasoconstrictive signal). Using in vitro explant assays coupled with pharmacological inhibition in mouse models of nerve injury, the authors demonstrate that ET-1/ETBR curbs axonal growth, and the ETAR/ETBR antagonist Bosentan boosts regrowth during the early phase of repair. In addition, Bosentan restores the ability of aged DRG neurons to regrow after nerve lesions. Despite Bosentan inhibiting both endothelin receptors A and B, comparison with an ETAR-specific antagonist indicates that the effects can be attributed to the ET-1/ETBR pathway. In the DRGs, ETBR is mostly expressed by SGCs (and a subset of Schwann cells) a cell type that previous studies, including work from this group, have implicated in nerve regeneration. SGCs ensheath and couple with DRG neurons through gap junctions formed by Cx43. Based on their own findings and evidence from the literature, the pro-regenerative effects of ETBR inhibition are in part attributed to an increase in Cx43 levels, which are expected to enhance neuron-SGC coupling. Finally, gene expression analysis in adult vs aged DRGs predicts a decrease in fatty acid and cholesterol metabolism, for which previous work by the authors has shown a requirement in SGCs to promote axon regeneration.Strengths:The study is well-executed and the main conclusion that "ETBR signaling inhibits axon regeneration after nerve injury and plays a role in age-related decline in regenerative capacity" (line 77) is supported by the data. Given that Bosentan is an FDA-approved drug, the findings may have therapeutic value in clinical settings where peripheral nerve regeneration is suboptimal or largely impaired, as it often happens in aged individuals. In addition, the study highlights the importance of vascular signals in nerve regeneration, a topic that has gained traction in recent years. Importantly, these results further emphasize the contribution of longneglected SGCs to nerve tissue homeostasis and repair. Although the study does not reach a complete mechanistic understanding, the results are robust and are expected to attract the interest of a broader readership.

We thank the reviewer for the positive comments, especially in regard to the rigor and originality of our study.

Weaknesses:Despite these positive comments provided above, the following points should be considered:(1) This study examines the contribution of the ET-1 pathway in the ganglia, and in vitro assays are consistent with the idea that important signaling events take place there. Nevertheless, it remains to be determined whether the accelerated axon regrowth observed in vivo depends also on cellular crosstalk mediated by ET-1 at the lesion site. Are ECs along the nerve secreting ET-1? What cells are present in the nerve stroma that could respond and participate in the repair process? Would these interactions be sensitive to Bosentan? It may be difficult to dissect this contribution, but it should at least be discussed.

We thank the reviewer for this important point and agree that the in vivo effects observed cannot rule out the contribution of ECs or SCs at the lesion site in the nerve. Dissecting the contribution of ETBR expressing cells in the nerve would require cell-specific manipulations that go beyond the scope of this manuscript. We have revised the Discussion section to highlight the potential contribution of ECs, fibroblast and SCs in the nerve.

(2) It is suggested that the permeability of DRG vessels may facilitate the release of "vascularderived signals" (lines 82-84). Is it possible that the ET-1/ETBR pathway modulates vascular permeability, and that this, in turn, contributes to the observed effects on regeneration?

We thank the reviewer for raising this interesting point. ET-1 can have an impact on vascular permeability. It was indeed shown that in high glucose conditions, increased trans-endothelial permeability is associated with increased Edn1, Ednra and Ednrb expression and augmented ET1 immunoreactivity (PMID: 10950122). It is thus possible that part of the effects observed results from altered vascular permeability. We have included this point in the Discussion section. Future experiments will be required to test how injury and age affects vascular permeability in the DRG.

(3) Is the affinity of ET-3 for ETBR similar to that of ET-1? Can it be excluded that ET-3 expressed by fibroblasts is relevant for controlling SGC responses upon injury/aging?

We thank the reviewer for raising this point. ET-1 binds to ETAR and ETBR with the same affinity, but ET3 shows a higher affinity to ETBR than to ETAR (Davenport et al. Pharmacol. Rev 2016 PMID: 26956245). We attempted to examine ET-3 level in adult and aged DRG by western blot, but in our hands the antibody did not work well enough, and we could not obtain clear results. We thus cannot exclude the possibility that ET-3 released by fibroblasts contribute to the effects we observe on axon regeneration. Indeed, in cultured cortical astrocytes, application of either ET-1 or ET-3 leads to inhibition of Cx43 expression. We have revised the text in the Discussion section to highlight the possibility that both ET-1 and ET-3 could participate on the ETBRdependent effect on axon regeneration.

(4) ETBR inhibition in dissociated (mixed) cultures uncovers the restraining activity of endothelin signaling on axon growth (Figure 2C). Since neurons do not express ET-1 receptors, based on scRNA-seq analysis, these results are interpreted as an indication that basal ETBR signaling in SGC curbs the axon growth potential of sensory neurons. For this to occur in dissociated cultures, however, one should assume that SGC-neuron association is present, similar to in vivo, or to whole DRG cultures (Figure 2C). Has this been tested?

We thank the reviewer for this point. In dissociated DRG culture, neurons, SGCs and other nonneuronal cells are present, but SGCs do not retain the surrounding morphology as they do in vivo. Within 24 hours in culture, SGCs lose their adhesive contacts with neuronal soma and adhere to the coverslip (PMID: 22032231, PMID: 27606776). We have included new data in Figure 2B to show that in our culture conditions, SGCs are present, but do not wrap neurons soma as they do in vivo. We also know from prior studies that the density of the culture affects axon growth, an effect that was attributed to trophic factors released from non-neuronal cells (Smith and Skene 1997). Therefore, although SGCs do not surround neurons, the signaling pathway downstream of ETBR may be present in culture and contribute to the release of trophic factors that influence axon growth. We have revised the Results section to better explain our in vitro results and their interpretation.

In both in vitro experimental settings (dissociated and whole DRG cultures) how is ETBR stimulated over up to 7 days of culture? In other words, where does endothelin come from in these cultures (which are unlikely to support EC/blood vessel growth)? Is it possible that the relevant ligand here derives from fibroblasts (see point #6)? Or does it suggest that ETBR can be constitutively active (i.e., endothelin-independent signaling)? Is there any chance that endothelin is present in the culture media or Matrigel?

We thank the reviewer for raising this point. Our single-cell data indicate that ET-1 is expressed by endothelial cells and ET-3 by fibroblasts. In dissociated DRG culture at 24h time point, all DRGs cells are present, including endothelial cells and fibroblasts, and could represent the source of ET-1 or ET-3. In the explant setting, it is also possible that both ET-1 and ET-3 are released by endothelial cells and fibroblasts during the 7 days in culture. According to information for the suppliers, endothelin is not present neither in the culture media nor in the Matrigel. While mutations can facilitate the constitutive activity of the ETBR receptor, we are not aware of data showing that endogenous ETBR can be constitutively active. Because the molecular mechanisms governing ETBR -mediated signaling remain incompletely understood (see for example PMID: 39043181, PMID: 39414992) future studies will be required to elucidate the detailed mechanisms activating ETBR in SGCs and its downstream signaling mechanisms. We have now expanded the Results and discussion sections to clarify these points.

(5) The discovery that ET-1/ETBR signaling in SGC curtails the growth capacity of axons at baseline raises questions about the physiological role of this pathway. What happens when ETBR signaling is prevented over a longer period of time? This could be addressed with pharmacological inhibitors, or better, with cell-specific knock-out mice. The experiments would certainly be of general interest, although not within the scope of this story. Nevertheless, it could be worth discussing the possibilities.

We agree that this is an interesting point. As mentioned above in response to point #1 of reviewer 1, the physiological role of this pathway could be to limit plasticity and prevent maladaptive neural rewiring that can happen after injury (Costigan et al 2009, PMID: 19400724), but can also hinder beneficial recovery after injury. Other mechanisms that limit axon regeneration capacity have been described and involve local mRNA translation and Rho signaling. We have revised the Discussion section to include these points. We agree that understanding the consequence of blocking ETBR over longer time periods is beyond the scope of the current study, but we now discuss the possibility that blocking ETBR with a cell specific KO approach could unravel its physiological function on target innervation and behavior.

(6) Assessing Cx43 levels by measuring the immunofluorescence signal (Figure 5E-F) is acceptable, particularly when the aim is to restrict the analysis to SGCs. The modulation of Cx43 expression by ET-1/ETBR plays an important part in the proposed model. Therefore, a complementary analysis of Cx43 expression by quantitative RT-PCR on sorted SGCs would be a valuable addition to the immunofluorescence data. Is this attainable?

We agree and have attempted to perform these types of experiments but encountered technical difficulties. We attempted to sorting SGCs from transgenic mice in which SGCs are fluorescently labeled. However, the cells did not survive the sorting process and died in culture. We think that increasing the viability of cells after sorting would require capillary- free fluorescent sorting approaches. However, we do not currently have access to such technology. We attempted this experiment with cultured SGCs, following a previously published protocol (Tonello et al. 2023 PMID: 38156033). In these experiments, SGCs are cultured for 8 days to obtain purity. We did not observe any difference in Cx43 protein or mRNA level upon treatment with ET-1 with or without BQ788. However, in these SGCs cultures, Cx43 displayed a diffuse localization, rather than puncta as observed in vivo. Therefore, despite our multiple attempts, quantifying Cx43 on sorted or purified SGCs was not attainable.

(7) The conclusions "We thus hypothesize that ETBR inhibition in SGCs contributes to axonal regeneration by increasing Cx43 levels, gap junction coupling or hemichannels and facilitating SGC-neuron communication" (lines 303-305) are consistent with the findings but seem in contrast with the effect of aging on gap junction coupling reported by others and cited in line 210: "the number of gap junctions and the dye coupling between these cells increases (Huang et al., 2006)". I am confused by what distinguishes a potential, and supposedly beneficial, increase in coupling after ETBR inhibition, from what is observed in aging.

We agree that the aging impact of Cx43 level and gap junction number appears contradictory. Procacci et al 2008 reported that Cx43 expression in SGCs decreases in the aged mice. Huang et al 2006 report that both the number of gap junctions and the dye coupling between these cells were found to increase with aging. Procacci et al suggested as a possible explanation for this apparent discrepancy that additional connexin types other than Cx43 may contribute to the gap junctions between SGCs in aged mice. Our snRNAseq data did not allow us to verify this hypothesis, because there were less SGCs in aged mice compared to adult, and connexin genes were detected in only 20% or less of SGCs. Furthermore, our quantification did not look specifically at gap junctions, but just at Cx43 puncta. Cx43 can also form hemichannels in addition to gap junctions, and can also perform non-channel functions, such as protein interaction, cell adhesion, and intracellular signaling. Thus, more research examining the role of Cx43 in SGCs is necessary to address this discrepancy in the literature. We have expanded the Discussion section to include these points.

(8) I find it difficult to reconcile the results in Figure 5F with the proposed model since (1) injury increases Cx43 levels in both adult and aged mice, (2) the injured aged/vehicle group has a similar level to the uninjured adult group, (3) upon injury, aged+Bosentan is much lower than adult+Bosentan (significance not tested). It seems hard to explain the effect of Bosentan only through the modulation of Cx43 levels. Whether the increase in Cx43 levels following ETBR inhibition actually results in higher SGC-neuron coupling has not been assessed experimentally.

We thank the reviewer for this point and agree that the effect of Bosentan is likely not exclusively through the modulation of Cx43 levels in SGCs, and that Cx43 levels may simply correlate with axon regenerative capacity. We have revised the manuscript to clarify this point. We have also added the missing significance test in Figure 5F.

Cell specific KO of Cx43 and ETBR would allow to test this hypothesis directly but is beyond the scope of the current study. We have not tested SGCs-neuron coupling, as these experiments are currently beyond our area of expertise. Cx43 has also other functions beyond gap junction coupling, such as protein interaction, cell adhesion, and intracellular signaling. Investigating the precise function of Cx43 would require in depth biochemical and cell specific experiments that are beyond the scope of this study. Furthermore, as we now mentioned in response to reviewer #2 point 5, ETBR signaling may also have other downstream effects in SGCs, such as glutamate transporters expression, or affect other cells in the nerve during the regeneration process. We have revised the Discussion section to include these alternative mechanisms.

**Reviewer #3(Public Review):**
Summary:This manuscript suggests that inhibiting ETBR via the FDA-approved compound Bosentan can disrupt ET-1-ETBR signalling that they found detrimental to nerve regeneration, thus promoting repair after nerve injury in adult and aged mice.Strengths:(1) The clinical need to identify molecular and cellular mechanisms that can be targeted to improve repair after nerve injury.(2) The proposed mechanism is interesting.(3) The methodology is sound.

We thank the reviewer for highlighting the strengths of our study

Weaknesses:(1) The data appear preliminary and the story appears incomplete.

We appreciate the reviewer’s point. We would like to emphasize that our results provide compelling evidence that ETBR signaling is a default brake on axon growth, and inhibiting this pathway promotes axon regeneration after nerve injury and counters the decline in regenerative capacity that occurs during aging. We also provide evidence that ETBR signaling regulates the levels of Cx43 in SGCs. Furthermore, our results document the use of an FDA approved compound to increase axon regeneration may be of interest to the broader readership, as there is currently no therapies to improve or accelerate nerve repair after injury. We agree that the detailed mechanisms operating downstream of ETBR will need to be elucidated. Answering these questions would first require cell specific KO of ETBR and Cx43 to confirm that this pathway is operating in SGCs to control axon regeneration. We would also need to identify how SGCs communicate with neurons to regulate axon regeneration, which is a large area of ongoing research that remains poorly understood. This extensive and highly complex set of experiments is beyond the scope of the current study. As we discussed in our response to reviewer #1 and #2 we attempted to perform numerous additional experiments to better define the role of ETBR signaling in SGCs in aging and have included additional results in Fig. 2B, Fig 3G-H, Fig 5A-E, and Figure 4- Figure Supplement 1and Figure 5- Figure Supplement 1. We have expanded the

Discussion to acknowledge the limitation of our study and to discuss possible mechanisms.

(2) Lack of causality and clear cellular and molecular mechanism. There are also some loose ends such as the role of connexin 43 in SGCs: how is it related to ET-1- ETBR signalling?

We thank the reviewer for this point and agree that the molecular mechanisms downstream of ETBR remain to be elucidated. However, we believe that our manuscript reports an interesting potential of an FDA-approved compound in promoting nerve repair. We focused on Cx43 downstream of ETBR signaling because decreased Cx43 expression in SGCs in ageing was previously established, but the mechanisms were not elucidated. Furthermore, it was reported that ET1 signaling in cultured astrocytes, which share functional similarities with SGCs, leads to the closure of gap junctions and reduction in Cx43 expression. Our study thus provides a mechanism by which ETBR signaling in SGCs regulates Cx43 expression. Whether Cx43 directly impact axon regeneration remains to be tested. Cell specific KO of Cx43 and ETBR would be required to answer this question. We have revised the Introduction and Discussion section extensively to provide a link between ETBR and Cx43 and to acknowledge the lack of causality in Cx43 in SGCs, as well as to provide additional potential mechanisms by which ETBR inhibition may promote nerve repair.

**Reviewer #2 (Recommendations For The Authors):**
In addition to the points listed in the Public Review section, please consider the following comments:(1) ETAR, which is high in mural cells, does not seem to be implicated in the reported proregenerative effects. Even so, can vasoconstriction be ruled out as an underlying cause of the age-dependent decline in axon regrowth potential and, more generally, in the effects of ET-1 inhibition on regeneration? This could be discussed.

We agree that we can’t exclude a role in vasoconstriction or effect on vascular permeability in the age-dependent decline in axon regrowth potential. However, our in vitro and ex vivo experiments, in which vascular related mechanisms are unlikely, suggest that vasoconstriction may not be a major contributor to the effects we observed.

(2) The manuscript (e.g. line 287-288) would benefit from a discussion of the role that blood vessels play in the peripheral nervous system, and possibly CNS, repair. Vessels were shown to accompany regenerating fibers and instruct the reorganization of the nerve tissue to favor repair potentially through the release of pro-regenerative signals acting on stromal cells, glia, and other cellular components. Highlighting these processes will help put the current findings into perspective.

We agree and have revised the Discussion section to better explain the role of blood vessels in orientating Schwann cells migration and guiding axon regeneration.

(3) The vast majority of the cells that are sequenced and shown in the UMAP in Figure 1C are from adult (3-month-old) mice [16,923 out of 18,098]. It would be useful to include the UMAP split (or color-coded) by timepoint to appreciate changes in cell clustering that may occur with aging.

We apologize for this misunderstanding, Figure 1C had all cells from all ages. However, the number of cells we obtained from the age group was insufficient to perform in depth analysis of each cell type. We have thus revised this section and Figure 1, now only presenting the data from adult mice.

It is not discussed why fewer cells were sequenced at later stages. Additionally, I do not know how to interpret the double asterisks next to the labeling "18,098 samples" in Figure 1C.

Since our original sequencing of adult and aged mice using 10x yielded so few cells from the aged DRG, we tested and optimized a new technology for single cell preparation of DRG using Illumina Single Cell 3’ RNA Prep. This preparation creates templated emulsions using a vortex mixer to capture and barcode single-cell mRNA instead of a microfluidics system. This method yielded much better results for nuclei recovery from aged DRG, with more nuclei and better quality of nuclei. Thus, we now present in Figure 5 and Figure 5- Figure Supplement 1 the results from snRNA-sequencing of aged and adult DRG using the Illumina single cell kit. The results of the snRNA-sequencing show a decreased abundance of SGCs in aged mice, consistent with the results from our morphology analysis with EM. We were also able to perform SGCs-specific pathway analysis because of the increased number of nuclei captured in the aged SGCs, which we included in the manuscript.

(4) The in vivo studies are designed to examine the effects of ETBR inhibition during the first phase of axon regrowth after nerve injury (1-3 days post-injury, dpi). Is there a reason why later stages have not been studied? It would be interesting to understand whether ETBR inhibition improves long-term recovery or is only effective at boosting the initial growth of axons through the lesion. It is possible that early inhibition will be enough for long-term recovery. If so, these experiments would define a sensitivity window with therapeutic value.

We agree that assessing functional recovery requires proper behavioral tests or morphological evaluations of reinnervation. To determine if Bosentan treatment has long-term effects on recovery, we administered Bosentan or vehicle for 3 weeks (daily for 1 week, and then once a week for the subsequent 2 weeks) after sciatic nerve crush. At 24 days after SNC, we assessed intraepidermal nerve fiber density (IENFD) in the injured paw and saw a trend towards increased fibers/mm in the treated animals (new Figure 3G,H). Future studies will examine how long-term Bosentan treatment affects functional recovery and innervation at later time points. Additionally, behavior assays will be needed to determine if these morphological changes relate to behavioral improvements using IENFD and behavior assays.

(5) I am unsure if the gene expression analysis shown in Figure 6 fits well into this story. It is interesting per se and in line with previous work from this group showing the relevance of fatty acid metabolism in SGCs for axon regeneration. Nevertheless, without a mechanistic link to endothelin signaling and Cx43/gap junction modulation, the observations derived from DEG analysis are not well integrated with the rest and may be more distracting than helpful. One limitation is that there is no cell-type information for the DEGs due to the small number of cells recovered from aged mice. For instance, if ETBR inhibition rescued gene downregulation associated with fatty acid/cholesterol metabolism, then the DGE results would become more relevant for understanding the cellular basis of the pro-regenerative effect, which at this point remains quite speculative (lines 264-265; lines 318-319).

We agree and have added new snRNA sequencing data to replace these findings see above response to point #4, new Figure 5 and Figure 5- Figure Supplement 1. The new data shows a decreased abundance of SGCs in aged mice, consistent with our TEM results. Pathway analysis revealed that aging triggers extensive transcriptional reprogramming in SGCs, reflecting heightened demands for structural integrity, cell junction remodeling, and glia–neuron interactions within the aged DRG microenvironment.

(6) It would be interesting to determine whether Bosentan increases SGC coverage of neuronal cell bodies in aged mice (Figures 6A-C).

We agree that this would be very interesting, but will require extensive EM analysis at different time points and is beyond the scope of the current manuscript.

(7) Finally, adding a summary model would help the readers.

We agree and have made a summary model, now presented in Figure 6F.

**Reviewer #3 (Recommendations For The Authors):**
Longer time points post-injury and assessment of functional recovery after Bosentan would be of great value here.

We agree that assessing functional recovery requires proper behavioral tests or morphological evaluations of reinnervation. To determine if Bosentan treatment has long-term effects on recovery, we administered Bosentan or vehicle for 3 weeks (daily for 1 week, and then once a week for the subsequent 2 weeks) after sciatic nerve crush. At 24 days after SNC, we assessed intraepidermal nerve fiber density in the injured paw and saw a trend towards increased fibers/mm in the treated animals (Fig 3). While the results do not reach significance, we decided to include this new data as it provides evidence that Bosentan treatment may also improves long term recovery. Future studies will be required examine how long-term Bosentan treatment affects functional recovery and innervation at later time points. Additionally, behavior assays will be needed to determine if these morphological changes relate to behavioral improvements.

It would be important to know how ET-1- ETBR signalling axis promotes the regeneration of axons:this remains unaddressed. What are the cells that are specifically involved? Endothelial cellsSGC- neurons- SC? There are no experiments addressing the role of any of these?

We agree that the molecular and cellular mechanisms by which ETBR signaling in SGCs promote axon regeneration remains to be elucidated. Answering these questions would first require cell specific KO of ETBR and Cx43 to confirm that this pathway is operating in SGCs to control axon regeneration. We would also need to identify how SGCs communicate with neurons to regulate axon regeneration, which is a large area of ongoing research that remains poorly understood. While these are important experiments, because of numerous technical and temporal constrains, we believe they are beyond the scope of the current manuscript.

How does connexin 43 in SGCs related to ET-1- ETBR signalling?

The relation between connexin 43 and ETBR signaling stems from observations made in astrocytes. ET1 signaling in cultured astrocytes, which share functional similarities with SGCs, was shown to lead to the closure of gap junctions and the reduction in Cx43 expression. Because Cx43 expression, a major connexin expressed in SGCs as in astrocytes, was previously shown to be reduced at the protein level in SGCs from aged mice, we decided to explore it this ETBR-Cx43 mechanism also operates in SGCs. We have revised the Introduction and Discussion section extensively to acknowledge the lack of causality in Cx43 expression SGCs and to provide additional potential mechanisms by which ETBR inhibition may promote nerve repair.